# Deterioration of postural control due to the increase of similarity between center of pressure and smooth-pursuit eye movements during standing on one leg

Hikaru Nakahara, Rukia Nawata©, Ryota Matsuo©, Tomohiro Ohgomori©*

Department of Rehabilitation, Osaka Kawasaki Rehabilitation University, Kaizuka, Osaka, Japan

© These authors contributed equally to this work.
* ohgomorit@kawasakigakuen.ac.jp

## Abstract

Upright postural control is regulated by afferent and efferent/reafferent visual mechanisms. There are two types of efferent and conjugate eye movements: saccades and smooth pursuits. Although postural control is improved by saccades, the effects of smooth pursuits on postural control are still debated, because the difficulties of postural and visual tasks differ in the previous research. Additionally, the mechanisms that interfere with postural control and smooth pursuit are not fully understood. To address these issues, we examined the effects of different patterns of smooth-pursuit eye movement on the path length of the center of pressure (COP) displacement under bipedal and unipedal standing conditions. The relative frequency and amplitude of the COP displacement were remarkably increased when uniform linear visual targets were presented during unipedal standing. In addition, dynamic time warping analysis demonstrated that the similarity between the displacement of the COP and eye movements was increased by the presentation of uniform linear visual targets with orientation selectivity during unipedal standing but not during bipedal standing. In contrast, the attenuation of similarity between the displacement of the COP and eye movements significantly decreased the path length, relative frequency, and amplitude of the COP displacement. Our results indicate that postural stability is deteriorated by the increase of similarity between the displacement of the COP and smooth-pursuit eye movements under unstable conditions.

## Introduction

Various types of afferent inputs from the external environment, such as visual, auditory, somatosensory, and proprioceptive contribute to postural control [1]. In particular, the impact of afferent visual inputs on postural control has been extensively studied. For instance, the path length and sway velocity of the center of pressure (COP) are greater when the eyes are closed than when they are open [2, 3]. Postural sway is larger in patients with glaucoma than

**Data Availability Statement:** All relevant data are within the manuscript and its Supporting information files.

**Funding:** This work was supported in part by the Grants-in-Aid for Scientific Research (KAKENHI) from the Japan Society for the Promotion of Science (20K07738 to T.O.) and the Research Foundation for Dementia of Osaka. The funders had no role in study design, data collection and analysis, decision to publish, or preparation of the manuscript.

**Competing interests:** The authors have declared that no competing interests exist.

in healthy controls according to visual field deficits [4]. In addition, the path length of postural sway was increased by visual field occlusion in healthy adults using a custom contact lens [5]. Interestingly, it has also been reported that the increase in COP speed caused by eye closing is greater in unipedal (UP) standing than in bipedal (BP) standing, regardless of sports expertise [6]. These reports indicate that afferent visual inputs strongly interfere with postural control.

Afferent inputs, as well as efferent/reafferent visual mechanisms, such as eye movements, influence postural stabilization. There are two main types of efferent and conjugate eye movements: saccades and smooth pursuits [7]. The sway area, amplitude of the COP displacement, and root mean square have been reported to be decreased by saccadic eye movements along both the anteroposterior (AP) and mediolateral (ML) directions in healthy young adults [8–10]. Postural stabilization is thought to be induced by a reduction in the mean amplitude and an increase in the mean frequency during high-frequency saccades in young adults [11]. However, changes in postural sway with smooth pursuit remain debated. For instance, the root mean square of COP displacement is increased by oscillating backgrounds and slow smooth pursuits [10, 12]. In contrast, smooth pursuits have been reported to significantly attenuate body sway, similar to saccades [13]. These controversial results strongly indicate that the reasons for the influence of smooth-pursuit eye movements on postural control have not been clarified.

In this study, we developed two hypotheses about the controversial interference between smooth-pursuit eye movements and postural control. First, the similarity between displacement of the COP and eye movements is important for postural control. Indeed, the observation of a swinging pendulum produced an increase in the lateral sway of the body [14]. In addition, the visual target moving in a continuous horizontal path deteriorated balance maintenance [15]. Moreover, postural instability differed during the smooth-pursuit eye movement tests in the horizontal, vertical, and diagonal directions under the same stance [16]. Second, the type of stance is critical for the impact of smooth pursuits on postural control. Body sway was attenuated when rectilinear and uniformly moving visual targets were presented during BP standing on a normal platform (an easy postural control task) [13]. By contrast, there was greater postural sway when similar visual targets were presented during a narrow stance and standing on a force plate covered with a foam cushion (difficult postural task) [17, 18]. To verify these hypotheses, several patterns of moving visual targets were presented to the participants under BP and UP standing conditions. Moreover, the similarity between the displacement of the COP and eye movements was examined to clarify the interfering mechanisms of smooth-pursuit eye movements with postural control.

## Methods

### Participants

In this study, 14 young male adults (age = 20.7±0.47 years old, height = 171.0±5.7 cm, body weight = 63.5±8.3 kg) were recruited. We determined pilot trial sample sizes as standardized effect sizes based on a previous report [19]. The research subjects were healthy and had no history of orthopedic and neurological diseases. In addition, no visual impairments were observed. Visual acuity was separately tested in each eye using Landolt C chart in a random order. The participants had a visual acuity of ≥ 1.0, with their glasses on or with the naked eye. All experiments were conducted in accordance with the Code of Ethics of the World Medical Association (Declaration of Helsinki) and were approved by the Ethics Committee of Osaka Kawasaki Rehabilitation University (OKRU20-A014). To avoid assentation, the results were not communicated to participants until the measurement schedule was completed.

## Experimental procedures

All experiments were conducted under standing conditions in a shaded space (Fig 1A). Each participant took part in 60 s of trials with a 2 min break after each trial. To offset the impact of the order of postural tasks, the participants were arbitrarily divided into two groups (group A, $n = 7$; group B, $n = 7$). Participants in one group were subjected to eight trials (2×postural tasks without moving visual target (WO), 2×postural tasks with randomly presented visual target (RM), 2×postural tasks with uniform linear visual target (SH), and 2×postural tasks with regular enlarged visual target (EL)) in BP standing condition. After one week, the same trials were conducted in the UP standing condition. In the other group, eight trials were conducted (2×postural tasks without moving the visual target, 2×postural tasks with randomly presented visual targets, 2×postural tasks with uniform linear visual targets, and 2×postural tasks with regular enlarged visual targets) in UP standing condition. After one week, the same trials were conducted in BP standing condition. The order of the trials was randomized. One week after completing the four types of visual tracking tasks during both BP and UP standing, an additional visual target was presented to the same participants during UP standing to reveal the influence of similarity between the displacement of the COP and eye movements on the COP displacement. The additional visual targets turn in unpredictable directions at the center of the monitor (ulSH). The path length, relative frequency, and amplitude of the COP displacement under the presentation of the ulSH-type visual targets was compared to that under the presentation of the SH-type visual targets. There are two reasons why participants were tested 1 week apart. One is the scheduling constraint of participants. Second is to eliminate the effect of fatigue on the COP displacement due to repeated measurement under the standing conditions based on the previous reports [20]. The participants were instructed to prioritize tracking the

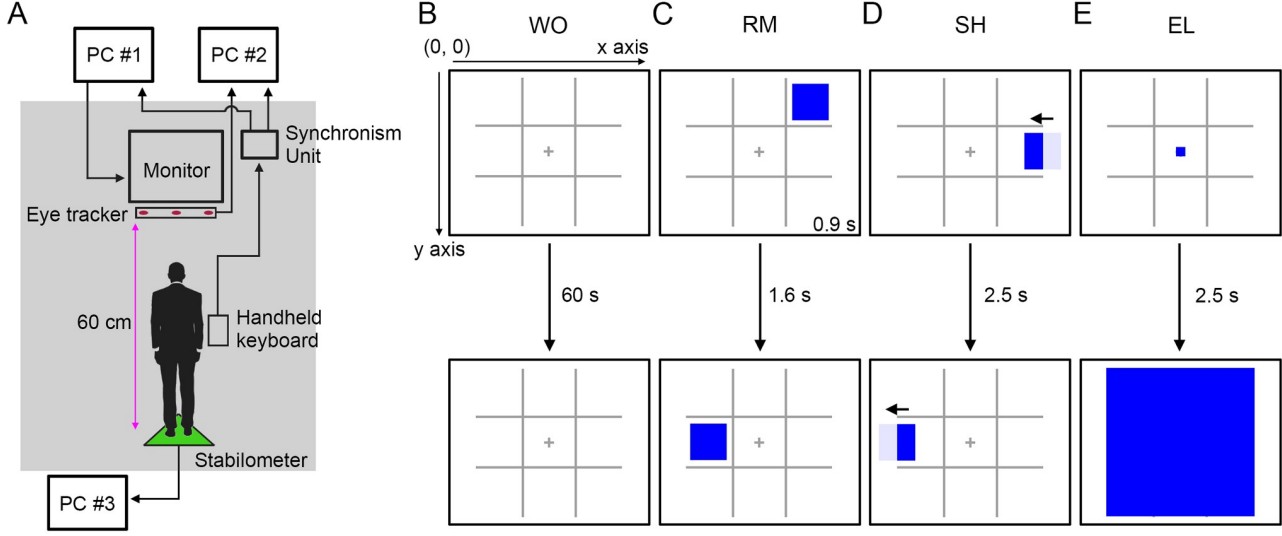

**Fig 1. Schematic representation of the COP movement and gaze point measurement.** (**A**) The measurement system of the COP movement and gaze point. Participants stood at the center of the stabilometer and had a handheld keyboard. The monitor was set 60 cm in front of a stabilometer at eye level. (**B-E**) The visual targets were presented on the monitor. A 3 × 3 grid were continuously presented on the monitor. In the case of WO, there were no moving signals for 60 s (B). In the case of RM, a blue square was presented at one of the 3 × 3 grid for 0.9 s. After the blank phase for 1.6 s, a blue square was represented at one of the 3 × 3 grid for 0.9 s. The flushing presentation of the blue square was repeated 24 times. The order of presented areas of blue squares was randomized (C). In the case of SH, the movement of blue square was linear at 12.6 °/s along the x-axis. A New blue square appeared from the right edge after the complete disappearance to left. A total of 24 blue squares were presented (D). In the case of EL, a small blue square (1 pixel × 1 pixel) was presented at the center of the monitor; it was enlarged into 1080 pixels × 1080 pixels for 2.5 s. A total of 24 small blue squares were presented and enlarged (E).

center position of the moving signals. In addition, participants started the presentation of visual targets simultaneously using a handheld keyboard (PCsensor, Guangdong, China) during both BP and UP standing. In the case of UP standing, the participants started the presentation of visual targets when they got their feet off the stabilometer.

## Postural control tasks

Postural control tasks were performed in two different standing patterns: BP and UP standing with eyes open. During BP standing, the participants stood at the center of the stabilometer (UM-BARII, Unimec, Tokyo, Japan) with their heels aligned and their toes pointing forward. In UP standing, the dominant foot was placed as a support according to individual kicking preference on the stabilometer, and the hip and knee of the lifted leg were flexed at 45 ˚. They crossed their arms in front of their bodies to avoid balance using their upper limbs and traced the signals presented on the monitor with their eyes. The sampling rate was 100 Hz.

## Visual targets

Visual targets were presented on a 27-inch monitor (1920 × 1080 pixels) set 60 cm in front of the stabilometer at eye level using Microsoft PowerPoint 2013. A 3 × 3 grid and the center position of the monitor were always presented. The grid was located at the center of the screen and marked with thin grey horizontal and vertical lines. A fixation cross is marked in the middle of the grid. The height and width of the grid were 360 pixels. It was previously reported that the observation of a swinging pendulum produced an increase in the lateral sway of the body [14]. Additionally, vertically and horizontally moving visual targets increased the power spectrum density of the COP displacement in the AP and ML directions, respectively [17]. Therefore, four types of moving visual target were used. In the case of WO, there were no moving visual targets (Fig 1B). For the RM, a blue square (300 × 300 pixels) was presented in a 3 × 3 grid for 0.9 s. After the blank phase for 1.6 s, a blue square was displayed again in the 3 × 3 grid (Fig 1C). The blue square was presented 24 times in the randomized areas. The RM-type visual task mainly induced the saccadic eye movement and rarely induced the smooth pursuit. In the case of SH, a blue square (300 × 300 pixels) is presented at the right edge of the grey-colored grid area. We made the blue square move linearly to the left (12.6 ˚/s) and disappear at the left edge of the grey-colored grid area (Fig 1D). The blue square was presented 24 times. The SH-type visual task mainly induced the smooth-pursuit eye movement and partially induced the saccade, when the eyes were moved from the left side of the screen to the right. In the case of EL, a blue square (1 × 1 pixel) appeared at the center of the monitor, which was enlarged 1080 times (Fig 1E). To reveal the significance of the similarity between gaze and COP movements, we used the ulSH-type visual target. In the case of ulSH, a blue square (300 × 300 pixels) is presented at the right edge of the grey-colored grid area. We then made the blue square move linearly (12.6 ˚/s) and turned toward unpredictable directions at the center of the grey-colored grid area. The total moving distance of the ulSH-type visual target is set to the same value as that of the SH-type target.

## Fast Fourier transformation

Prior to calculating amplitude, we examined the distributions of standard deviation (SD) of the COP displacement in the AP and ML directions (S1 Fig). There were several data which was more than 1.5 interquartile ranges below the first quartile or above the third quartile. However, these data could not be designated as outliers due to the small sample size in present study. Therefore, the temporal data in the AP and ML directions obtained from all participants during BP and UP standing (60 s) were changed to frequencies using Bluestein's fast Fourier

transformations, as reported previously [21]. These signals were low pass filtered with a cut-off at 3 Hz based on the previous report [22]. In addition, the power spectrum was divided into three frequency bandwidths: low (0–0.3 Hz), middle (0.3–1.0 Hz), and high (1.0–3.0 Hz) [23]. The relative proportion of the area under the spectral plots of power in each frequency bandwidth was calculated. The sum of the area under the spectral plots of amplitude in each frequency bandwidth was designated as the amplitude of the COP displacement.

### Eye-tracking

Tobii eye tracker 5 (Tobii Technology K. K., Stockholm, Sweden) was attached under a 27-inch monitor set in a shaded space. Calibration was performed using an automatically installed Tobii Experience software (Tobii Technology K. K.). The gaze point was changed to the position of the cursor using Miyasuku EyeConLT2 (Unicorn Corp., Hiroshima, Japan). The temporal coordinates of the cursor were recorded as text files in Python (JetBrains self-regulatory organization, Prague, Czech Republic). The sampling rate was 100 Hz. The recording of the gaze point and presentation of visual targets were concurrently started using a four-port USB synchronous controller connected to a handheld keyboard (PCsensor). When the subject was standing with the 27-inch monitor set at a 60 cm distance in front, the measurement error of the eye-tracking method was approximately 2 ˚ in the x- and y-axes owing to a limitation of the measurement, which was primarily included in the central vision [24].

### Dynamic time warping analysis

There are several methods to measure the similarity between two time-series data, including cross-correlation and wavelet coherence analyses [25]. However, the frequency bandwidths of the COP displacement differ from those of eye movements. The dynamic time warping (DTW) method is possible to quantify the similarity of two time-series data with non-linear extension and contraction allowed, even though the frequency and the number of datasets are different. Therefore, we used the DTW analysis to measure the similarity between two temporal sequences; that is, the displacement of the COP and eye movement [26]. We minimized the distance between the two temporal sequences using the DTW package in R software without band filters (Sakoe-Chiba and Itakura), because it was impossible to estimate the suitable window size for matching the COP and eye movements [27]. The displacement of the COP in the AP and ML directions was compared with that of the gaze point in the x- and y-axes using the brute force method.

### Statistical analysis

Before statistical analysis, normality was checked using the Shapiro-Wilk W test in the R software. The data had non-parametric distribution. Data were statistically analyzed using Kaleidagraph 4.5 (Hulinks, Tokyo, Japan). The path length, relative frequency and amplitude of the COP displacement, and nearest dynamic time warping distance were the dependent variables, and the types of visual targets were the independent variables. Each group had 14 participants. We evaluated the statistical differences among the four conditions using Friedman's analysis of variance and multiple Wilcoxon signed-rank tests. We applied Bonferroni correction to perform multiple comparisons. To perform the Bonferroni correction, the critical $P$-value ($\alpha$) was divided by the number of comparisons (six comparisons). Differences were considered significant when a $P$-value < 0.00833 was obtained. By contrast, we evaluated the statistical differences between the two conditions using Wilcoxon signed-rank tests. Differences were considered significant when the $P$-value < 0.05 was obtained.

## Results

### The path length of the COP displacement was affected by the smooth-pursuit eye movements

We first examined the changes in the path length of the COP displacement after the presentation of visual targets during BP and UP standing (Fig 2). In general, the displacement of the COP was lower during BP standing without the moving target (WO), and there was no remarkable difference among the four types of visual targets (Fig 2B–2D). In contrast, the

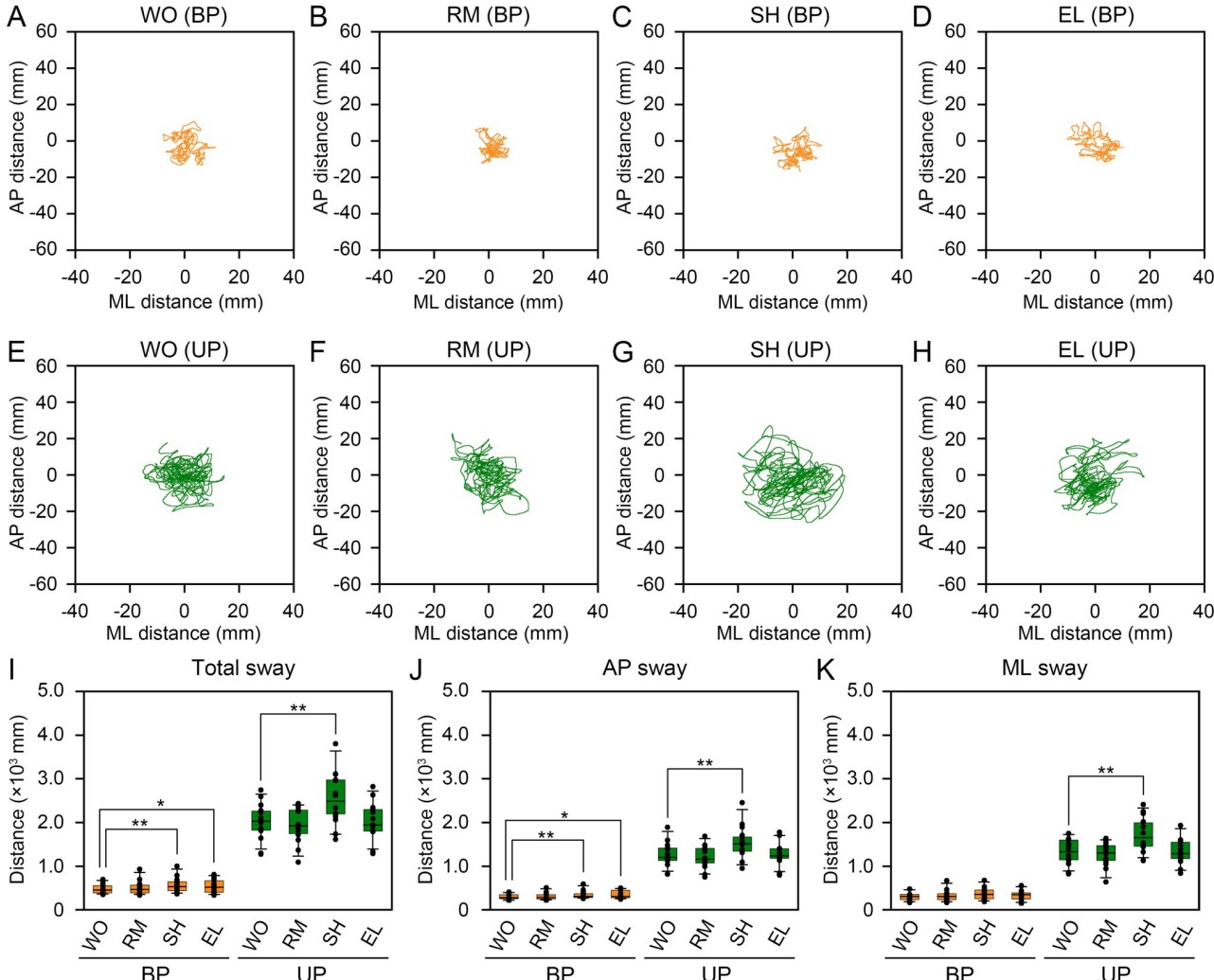

**Fig 2. Smooth-pursuit eye movement affected the path length of the COP displacement.** (**A-D**) Representative trajectories of the COP during bipedal (BP) standing under the presentation of no moving signals (WO, A), random flushing squares (RM, B), squares that shift from right to left (SH, C), and enlarged squares (EL, D). (**E-H**) Representative trajectories of the COP during unipedal (UP) standing under the presentation of the WO- (E), RM- (F), SH- (G), and EL-type (H) visual targets. (**I**) The total distances of COP movement under the presentation of the WO-, RM-, SH-, and EL-type visual targets during BP (orange) and UP (green) standing. (**J**) The distances of COP movement in the anteroposterior (AP) direction under the presentation of the WO-, RM-, SH-, and EL-type visual targets during BP (orange) and UP (green) standing. (**K**) The distances of COP movement in the mediolateral (ML) direction under the presentation of the WO-, RM-, SH-, and EL-type visual targets during BP (orange) and UP (green) standing. The box plots represent the median, first and third quartiles (boxes), and fifth and 95th percentiles (whiskers). The number of participants: $n = 14$. Statistical differences were analyzed using Friedman's analysis of variance followed by multiple Wilcoxon's signed-rank test with Bonferroni correction. Abbreviations: AP, anteroposterior; BP, bipedal; ML, mediolateral; UP, unipedal. Statistical significance is indicated by asterisks: * $P < 0.00833$, ** $P < 0.00167$.

displacement of the COP was greater during UP standing than during BP standing without the moving target (Fig 2E). The displacement of the COP did not remarkably differ between WO and RM during UP standing (Fig 2F). The displacement of the COP was larger in the SH than that in the WO during UP standing (Fig 2G). The displacement of the COP did not differ remarkably between WO and EL during UP standing (Fig 2H).

The total distance of COP movement was larger in SH and EL than in WO, and that in RM and WO was similar during BP standing ($\chi^2_{3,39}$ = 19.1, $P$ = 0.00026, Fig 2I, Table 1, S1 Table). The total distance of COP movement was also larger in the SH than in the WO, and the total distances of COP movement in the RM and EL were similar to those in the WO during UP standing ($\chi^2_{3,39}$ = 27.6, $P$ < 0.0001, Fig 2I, Table 1, S1 Table). The distance of COP movement in the AP direction was larger in SH and EL than in WO, and that in RM and WO was similar during BP standing ($\chi^2_{3,39}$ = 15.7, $P$ = 0.00132, Fig 2J, Table 1, S1 Table). During UP standing, the distance of COP movement in the AP direction was larger in the SH than in the WO ($\chi^2_{3,39}$ = 28.9, $P$ < 0.0001, Fig 2J, Table 1, S1 Table). In contrast, the distance of COP movement in the ML direction did not significantly differ among four groups during BP standing ($\chi^2_{3,39}$ = 11.4, $P$ = 0.00975, Fig 2K, Table 1, S1 Table). During UP standing, the distance of COP movement in the ML direction was larger in the SH than in the WO, and the distance of COP movement in the ML direction in the RM and EL did not significantly differ from those in the WO ($\chi^2_{3,39}$ = 28.5, $P$ < 0.0001, Fig 2K, Table 1, S1 Table). These data indicate that COP movement activity was increased by the presentation of uniform linear visual targets during both the easy and difficult postural tasks.

## The impact of smooth-pursuit eye movement on the amplitude and frequency of COP displacement

Prior to changing to frequencies using fast Fourier transformation, the changes in SD of temporal displacement of the COP in the AP and ML directions were examined (S1 Fig). The SD in the AP direction remained unaltered among the four groups during BP standing ($\chi^2_{3,39}$ = 1.89, $P$ = 0.596, S1A Fig, S1 Table). The SD in the AP direction was slightly, but not significantly, altered ($\chi^2_{3,39}$ = 8.31, $P$ = 0.0399, S1A Fig, S1 Table, Table 1). The SD in the ML direction also remained unaltered among the four groups during BP standing ($\chi^2_{3,39}$ = 3.26, $P$ = 0.354, S1A Fig, S1 Table). By contrast, The SD in the ML direction was significantly increased by the presentation of SH-type visual target ($\chi^2_{3,39}$ = 20.9, $P$ = 0.00011, S1A Fig, S1 Table, Table 1). These data indicate that body sway was increased in the ML direction by the presentation of SH-type visual target. The change in the COP movement was mainly affected by two components: the relative proportion of the frequency bandwidths and amplitudes. Since the smooth-pursuit eye movement significantly increased the COP movement, we next examined the potential effects of smooth-pursuit eye movement on the relative proportion of frequency bandwidths and amplitude of COP displacement in the AP and ML directions during BP and UP standing (Fig 3). The power spectra were calculated from temporal coordinates in the AP and ML directions. First, we examined the relative proportions of the frequency bandwidths of COP displacement. The relative proportions of postural sway in low- (0.1–0.3 Hz, $\chi^2_{3,39}$ = 2.49, $P$ = 0.478), middle- (0.3–1.0 Hz, $\chi^2_{3,39}$ = 2.66, $P$ = 0.448), and high- (1.0–3.0 Hz, $\chi^2_{3,39}$ = 3.52, $P$ = 0.318) frequency bandwidths remained unaltered among the four groups in the AP direction during BP standing (Fig 3A, S1 Table). Consistent with our previous report, the relative proportion of high-frequency bandwidth increased during UP standing compared with BP standing (Fig 3B, S1 Table). The relative proportion of postural sway in the low-frequency bandwidth was slightly, but not significantly, decreased ($\chi^2_{3,39}$ = 11.9, $P$ = 0.00768, Fig 3B, Table 1, S1 Table), and that in the middle-frequency bandwidth was

**Table 1. Summary of multiple Wilcoxon's test.**

| | | WO vs. RM | | WO vs. SH | | WO vs. EL | |
|---|---|---|---|---|---|---|---|
| | | Z-value | P-value | Z-value | P-value | Z-value | P-value |
| Fig 2I | BP | -0.596 | 0.583 | -2.98 | 0.00122 | -2.79 | 0.00305* |
| | UP | -2.04 | 0.0419 | -3.30 | 0.000122 | -0.157 | 0.903 |
| Fig 2J | BP | -0.345 | 0.761 | -2.98 | 0.00122 | -2.86 | 0.00232* |
| | UP | -1.73 | 0.00906 | -3.3 | 0.000122 | -0.157 | 0.903 |
| Fig 2K | BP | -0.569 | 0.583 | -2.54 | 0.00855 | -2.54 | 0.00855 |
| | UP | -1.91 | 0.0580 | -3.3 | 0.000122 | -0.345 | 0.761 |
| Fig 3B | Low | -1.41 | 0.173 | -2.67 | 0.00916 | -1.98 | 0.0494 |
| | Mid | -1.54 | 0.135 | -2.86 | 0.00232 | -2.73 | 0.00403* |
| Fig 3G | Mid | -1.16 | 0.268 | -2.92 | 0.00171 | -1.66 | 0.104 |
| Fig 3H | High | -1.22 | 0.241 | -2.92 | 0.00171 | -0.722 | 0.502 |
| Fig 3J | Low | -0.0314 | 1.00 | -2.48 | 0.0132 | -0.722 | 0.502 |
| | Mid | -0.220 | 0.855 | -2.79 | 0.00305 | -0.973 | 0.358 |
| Fig 3N | Low | -1.35 | 0.194 | -2.61 | 0.0403 | -0.157 | 0.903 |
| Fig 3O | Mid | -1.41 | 0.173 | -3.30 | 0.000122 | -0.0942 | 0.952 |
| Fig 3P | High | -2.23 | 0.0245 | -3.30 | 0.000122 | -0.0314 | 1.00 |
| Fig 4F | BP | -1.73 | 0.0906 | -2.79 | 0.00305 | -0.0942 | 0.952 |
| Fig 4G | UP | -0.785 | 0.463 | -3.11 | 0.000610 | -1.10 | 0.296 |
| | | x-ML vs. x-AP | | x-ML vs. y-ML x-ML vs. y-AP | | | |
| | | Z-value | P-value | Z-value | P-value | Z-value | P-value |
| Fig 4H | | -3.23 | 0.000244 | -3.11 | 0.00061 | -1.35 | 0.194 |
| Fig 5E | | -2.61 | 0.00671 | -2.98 | 0.00122 | -2.35 | 0.0166 |
| S1A Fig | UP | -2.23 | 0.0245 | -1.04 | 0.326 | -1.41 | 0.173 |
| S1B Fig | UP | -0.973 | 0.358 | -3.04 | 0.000855 | -0.157 | 0.903 |

\* $P < 0.00833$,

\** $P < 0.00167$

increased by the presentation of the SH-type visual target in the AP direction ($\chi^2_{3,39} = 13.5$, $P = 0.00375$, Fig 3B, Table 1, S1 Table). The relative proportion of postural sway in the high-frequency bandwidth remained unaltered among the four groups in the AP direction ($\chi^2_{3,39} = 2.83$, $P = 0.419$, Fig 3B, S1 Table). We then calculated the amplitude spectra from the temporal coordinates of the COP data in the AP and ML directions. The amplitudes of the low- ($\chi^2_{3,39} = 1.54$, $P = 0.672$), middle- ($\chi^2_{3,39} = 13.1$, $P = 0.0044$), and high- ($\chi^2_{3,39} = 3.09$, $P = 0.379$) frequency bandwidths in the AP direction did not differ significantly among the four groups during BP standing (Fig 3C–3E, S1 Table). The amplitude of the low-frequency bandwidth in the AP direction did not differ significantly among the four groups during UP standing ($\chi^2_{3,39} = 2.83$, $P = 0.431$, Fig 3F, S1 Table). In contrast, the amplitudes of the middle- ($\chi^2_{3,39} = 22.5$, $P < 0.0001$, Fig 3G, S1 Table) and high- ($\chi^2_{3,39} = 22.0$, $P < 0.0001$, Fig 3H, S1 Table) frequency bandwidths in the AP direction were increased by the presentation of the SH-type visual target during UP standing (Table 1). Subsequently, we examined the relative proportions of the frequency bandwidths of COP displacement in the ML direction. The relative proportions of postural sway in the low- ($\chi^2_{3,39} = 0.6$, $P = 0.896$), middle- ($\chi^2_{3,39} = 0.6$, $P = 0.896$), and high- ($\chi^2_{3,39} = 2.14$, $P = 0.543$) frequency bandwidths were unaltered among the four groups in the ML direction during BP standing (Fig 3I, S1 Table). Consistent with the AP direction, the relative proportion of high-frequency bandwidth increased during UP standing compared with that

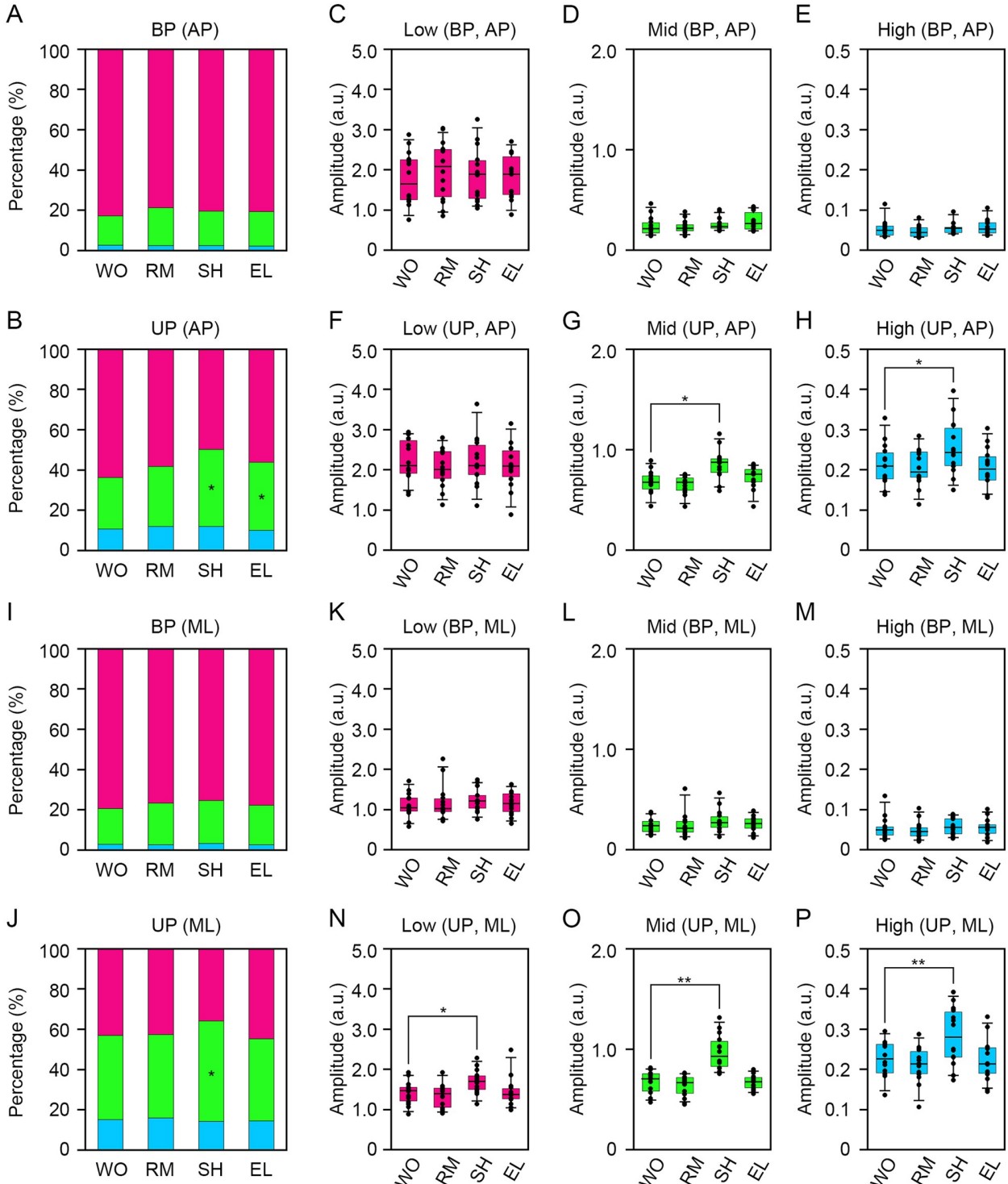

**Fig 3. Changes in the relative proportion and amplitude of three frequency bandwidths of the COP displacement by smooth-pursuit eye movement.** (**A, B**) Alterations in the relative proportions of postural sway in low- (0.1–0.3 Hz, magenta), middle- (0.3–1.0 Hz, green), and high- (1.0–3.0 Hz, cyan) frequency bandwidths in the anteroposterior (AP) direction under the presentation of no moving signals (WO), random flushing squares (RM), squares which shift from right to left (SH), and enlarged squares (EL) during bipedal (BP, A) and unipedal (UP, B) standing. (**C-E**) Alterations in the amplitudes of postural sway in low- (C), middle- (D), and high- (E) frequency bandwidths in the AP direction under the presentation of the WO-, RM-, SH-, and EL-type visual targets during BP standing. (**F-H**) Alterations in the amplitudes of postural sway in low- (F), middle- (G), and high- (H) frequency bandwidths in the AP direction under the presentation of the WO-, RM-, SH-, and EL-type visual targets during UP standing. (**I, J**) Alterations in the relative proportions of postural sway in low-, middle-, and high-frequency bandwidths in the

mediolateral (ML) direction under the presentation of the WO-, RM-, SH-, and EL-type visual targets during BP (I) and UP (J) standing. (**K-M**) Alterations in the relative proportions of postural sway in low- (K), middle- (L), and high- (M) frequency bandwidths in the mediolateral (ML) direction under the presentation of the WO-, RM-, SH-, and EL-type visual targets during BP standing. (**N-P**) Alterations in the amplitudes of postural sway in low- (N), middle- (O), and high- (P) frequency bandwidths in the ML direction under the presentation of the WO-, RM-, SH-, and EL-type visual targets during UP standing. The box plots represent the median, first and third quartiles (boxes), and fifth and 95th percentiles (whiskers). The number of participants: $n$ = 14. Statistical differences were analyzed using Friedman's analysis of variance followed by multiple Wilcoxon's signed-rank test with Bonferroni correction. Abbreviations: AP, anteroposterior; BP, bipedal; ML, mediolateral; UP, unipedal. Statistical significance is indicated by asterisks: $^{*}$ $P < 0.00833$, $^{**}$ $P < 0.00167$.

during BP standing (Fig 3J, S1 Table). The relative proportions of low- ($\chi^2$ $_{3,39}$ = 10.5, $P$ = 0.0145) and high- ($\chi^2$ $_{3,39}$ = 1.8, $P$ = 0.615) frequency bandwidths remained unaltered in the ML direction among the four groups (Fig 3J, Table 1, S1 Table). By contrast, the relative proportion of the middle-frequency bandwidth was increased by the presentation of the SH-type visual target ($\chi^2$ $_{3,39}$ = 15.4, $P$ = 0.00148, Fig 3J, Table 1, S1 Table). The amplitudes of low- ($\chi^2$ $_{3,39}$ = 3.09, $P$ = 0.379), middle- ($\chi^2$ $_{3,39}$ = 11.3, $P$ = 0.0101), and high- ($\chi^2$ $_{3,39}$ = 10.9, $P$ = 0.0124) frequency bandwidths in the ML direction did not significantly differ among the four groups during BP standing (Fig 3K–3M, S1 Table). The amplitudes of the low- ($\chi^2$ $_{3,39}$ = 11.1, $P$ = 0.0110), middle- ($\chi^2$ $_{3,39}$ = 25.8, $P < 0.0001$), and high- ($\chi^2$ $_{3,39}$ = 29.7, $P < 0.0001$) frequency bandwidths in the ML direction were significantly increased by the presentation of the SH-type visual target (Fig 3N–3P, Table 1, S1 Table). In contrast, these were not affected by the presentation of the RM- and EL-type visual targets (Fig 3N–3P, Table 1, S1 Table). These data indicate that the relative frequency and amplitude of COP displacement increased by the presentation of a uniform linear visual target during a difficult postural task.

## Similarity between the displacement of the COP and eye movements under the presentation of uniform linear visual targets

It was previously reported that horizontally moving visual targets increase the displacement of the COP in the lateral direction [14]. Next, we examined the similarity between the displacement of the COP and the eye movements (Fig 4). Generally, the displacement of the COP was small in the AP and ML directions during the presentation of the WO-type visual target (Fig 4A$_1$). The gaze points rarely moved along the x- and y-axes (Fig 4A$_2$). The displacement of the COP did not differ in the AP and ML directions during the presentation of the RM-type visual target compared to the WO-type target (Fig 4B$_1$). The gaze points were moved according to the presentation of visual targets on the x- and y-axes (Fig 4B$_2$). In contrast, the COP was remarkably displaced in the AP and ML directions by the presentation of the SH-type visual target (Fig 4C$_1$). The gaze point on the x-axis moved regularly, whereas it rarely moved along the y axis (Fig 4C$_2$). The displacement of the COP also did not differ in the AP and ML directions during the presentation of the EL-type visual target compared with the WO-type visual target (Fig 4D$_1$). The gaze points rarely moved along the x- and y-axes (Fig 4D$_2$). The similarity between the two waves was statistically evaluated using the DTW method (Fig 4E). The DTW distance was extended by the presentation of the SH-type visual target compared to that of the WO-type target during BP standing ($\chi^2$ $_{3,39}$ = 12.1, $P$ = 0.0071, Fig 4F, Table 1, S1 Table). In contrast, the DTW distance was significantly decreased by the presentation of the SH-type visual target compared with the WO-type visual target during UP standing ($\chi^2$ $_{3,39}$ = 15.9, $P$ = 0.00116, Fig 4G, Table 1, S1 Table). The orientation selectivity of the similarity between the displacement of the COP and eye movements was examined using the brute-force DTW method. The ratio of the DTW distance was especially lower when the COP displacements in the ML direction were compared with those of the gaze point along the x-axis ($\chi^2$ $_{3,39}$ = 21.9, $P < 0.0001$, Fig 4H, Table 1, S1 Table). These data indicate that similarity between the

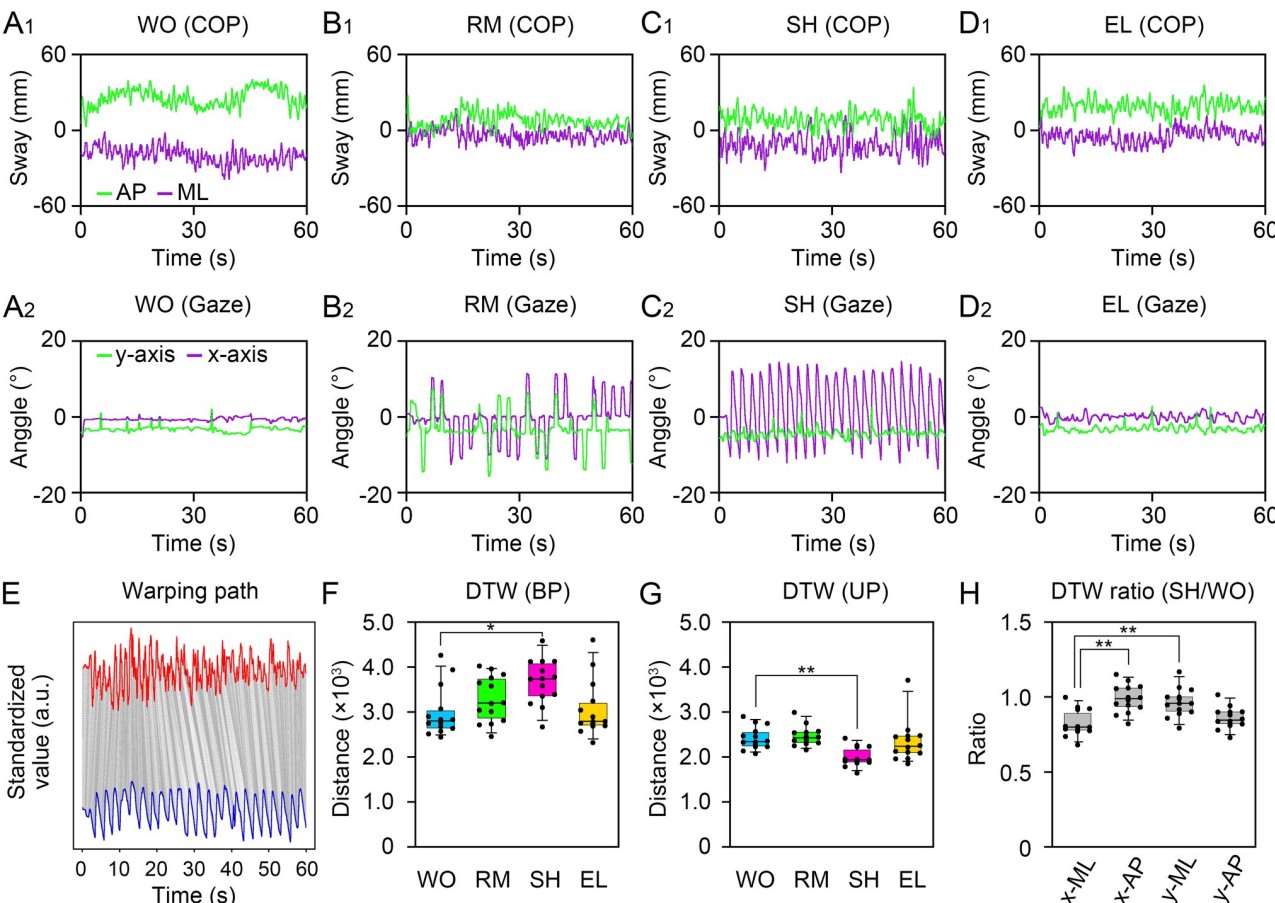

**Fig 4. The similarity between the displacement of the center of pressure (COP) and gaze point.** ($A_1$) Temporal displacement of the COP in the anteroposterior (AP, green) and mediolateral (ML, purple) directions under the presentation of the WO-type visual target during unipedal (UP) standing. ($A_2$) Temporal displacement of the gaze point in x- (purple) and y-axes (green) of the monitor under the presentation of the WO-type visual target during UP standing. ($B_1$) Temporal displacement of the COP in the AP and ML directions under the presentation of the RM-type visual target during UP standing. ($B_2$) Temporal displacement of the gaze point in the x- and y-axes of the monitor under the presentation of the RM-type visual target. ($C_1$) Temporal displacement of the COP in the AP and ML directions under the presentation of the SH-type visual target during UP standing. ($C_2$) Temporal displacement of the gaze point in the x- and y-axes of the monitor under the presentation of the SH-type visual target during UP standing. ($D_1$) Temporal displacement of the COP in the AP and ML directions under the presentation of the EL-type visual target during UP standing. ($D_2$) Temporal displacement of the gaze point in the x- and y-axes of the monitor under the presentation of the EL-type visual target during UP standing. (**E**) Representative warping alignment between temporal displacements of the COP in the ML direction (red) and the gaze point in the x-axis (blue) under the presentation of the SH-type visual target. (**F, G**) The nearest warping distance between the standardized plots of the COP and the gaze under the presentation of the WO- (cyan), RM- (green), SH- (magenta), and EL-type (yellow) visual targets during BP (F) and UP (G) standing. (**H**) The ratio of nearest warping distance (SH/WO) during UP standing. The box plots represent the median, first and third quartiles (boxes), and fifth and 95th percentiles (whiskers). The number of participants: $n = 14$. Statistical differences were analyzed using Friedman's analysis of variance followed by multiple Wilcoxon's signed-rank test with Bonferroni correction. Abbreviations: AP, anteroposterior; BP, bipedal; DTW, dynamic time warping; ML, mediolateral; UP, unipedal. Statistical significance is indicated by asterisks: * $P < 0.00833$, ** $P < 0.00167$.

displacement of the COP and eye movements was increased by the presentation of a uniform linear visual target with orientation selectivity during a difficult postural task.

## Inhibition of postural sway due to the decrease of similarity between the displacement of the COP and eye movements

Finally, we aimed to demonstrate the potential effects of similarity between the displacement of the COP and eye movements but not the movement distance of gaze on postural control. Thus, we examined the potential inhibitory effects of similarity between the displacement of

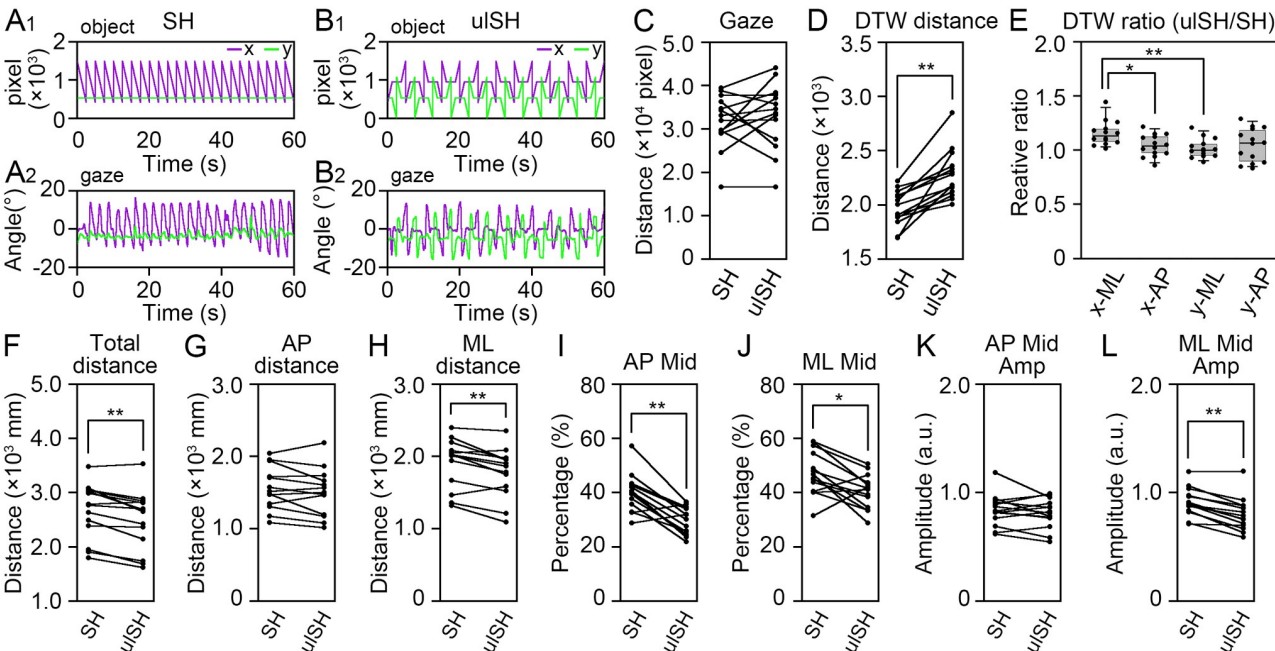

**Fig 5. The decrease of similarity between the displacement of the center of pressure (COP) and the gaze point inhibited the displacement of the COP. (A)** The temporal coordinates of predictable SH-type (SH) visual object (A$_1$) and gaze point (A$_2$) in the x- (purple) and y- (green) axes respectively. **(B)** The temporal coordinates of unpredictable SH-type (ulSH) visual object (B$_1$) and gaze point (B$_2$) in the x- and y-axes respectively. **(C)** The change in the total distance of gaze point under the presentation of the SH- and ulSH-type visual targets. **(D)** The nearest warping distance between the standardized plots of the COP and the gaze under the presentation of the SH- and ulSH-type visual targets. **(E)** The relative ratio of nearest warping distance (ulSH/SH) during UP standing. **(F)** The total distance of COP movement under the presentation of the SH- and ulSH-type visual targets. **(G, H)** The distance of COP movement in the anteroposterior (AP, G) and mediolateral (ML, H) directions under the presentation of the SH- and ulSH-type visual targets. **(I, J)** The relative percentage of middle-frequency bandwidth of postural sway in the AP (I) and ML (J) directions under the presentation of the SH- and ulSH-type visual targets. **(K, L)** The amplitude of postural sway in middle-frequency bandwidths in the AP (K) and ML (L) directions under the presentation of the SH- and ulSH-type visual targets during UP standing. The box plots represent the median, first and third quartiles (boxes), and fifth and 95th percentiles (whiskers). The number of participants: n = 14. Statistical differences were analyzed using Wilcoxon's signed-rank test. Abbreviations: AP, anteroposterior; BP, bipedal; DTW, dynamic time warping; ML, mediolateral; UP, unipedal. Statistical significance is indicated by asterisks: * P < 0.05, ** P < 0.01.

the COP and eye movements on postural sway using an additional visual target (Fig 5). The movement of the predictable visual target (Fig 5A$_1$) was similar to that of the gaze point along the x- and y-axes (Fig 5A$_2$). The movement of the unpredictable visual target (Fig 5B$_1$) was also closely associated with that of the gaze point along the x- and y-axes (Fig 5B$_2$). The total distance of eye movement did not differ significantly between the presentation of the SH- and ulSH-type visual targets (Z = -0.910, P = 0.402, Fig 5C, S1 Table). The DTW distance was significantly increased by the presentation of the ulSH-type visual target compared with the SH-type target during UP standing (Z = -3.30, P = 0.000122, Fig 5D, S1 Table). The orientation selectivity of the similarity between the COP displacement and eye movements was examined using the brute-force DTW method. The relative ratio of the DTW distance was larger when the COP displacements in the ML direction were compared with those of the gaze point along the x-axis ($\chi^2_{3,39}$ = 11.4, P = 0.00975, Fig 5E, Table 1, S1 Table). The total distance of COP movement was smaller for the ulSH-type visual target than for the SH-type target (Z = -2.98, P = 0.00122, Fig 5F, S1 Table). The path length of the COP displacement in the AP direction was unaltered by the presentation of the ulSH-type visual target (Z = -1.54, P = 0.135, Fig 5G, S1 Table). The path length of the COP displacement in the ML direction significantly decreased when the ulSH-type visual target was presented (Z = -2.92, P = 0.00171, Fig 5H, S1 Table). Prior to changing to frequencies using fast Fourier transformation, the change in SD of

temporal displacement of the COP in the AP and ML directions were examined (S1 Fig). The SD in the AP direction was increased by the presentation of ulSH-type visual target ($Z$ = -2.04, $P$ = 0.0419, S1C Fig). By contrast, the SD in the Ml direction did not differ between two groups ($Z$ = -1.35, $P$ = 0.194, S1D Fig). The relative percentages of the middle-frequency bandwidth in both the AP ($Z$ = -3.11, $P$ = 0.00061, Fig 5I, S1 Table) and ML ($Z$ = -2.42, $P$ = 0.0134, Fig 5J, S1 Table) directions were smaller for the ulSH-type visual target than for the SH-type. The amplitude of the middle-frequency bandwidth in the AP direction did not differ significantly between the presentation of the SH- and ulSH-type visual targets ($Z$ = -0.785, $P$ = 0.463, Fig 5K, S1 Table). In contrast, the amplitude of the middle-frequency bandwidth in the ML direction was significantly lower for the ulSH-type visual target than for the SH-type target ($Z$ = -3.23, $P$ = 0.00024, Fig 5L, S1 Table). These data indicate that the decrease of similarity between eye and COP movements attenuates the displacement of the COP.

## Discussion

### The effects of eye movement task on the postural control

It was previously reported that the saccadic eye movement significantly attenuated postural control [8, 9, 11]. In this study, we revealed that the path length of the COP displacement under the presentation of the RM-type visual target tend to be smaller than that under the presentation of the WO-type during UP standing ($P$-value = 0.0419 in a direct comparison between two groups) but not BP standing. In addition, the amplitude of the high-frequency bandwidth in the ML direction under the presentation of the RM-type visual target also tend to be smaller than that under the presentation of the WO-type ($P$-value = 0.0245 in a direct comparison between two groups) but not BP standing. However, the change in the path length of the COP displacement and the amplitude of high-frequency bandwidth in the ML direction were smaller under the presentation of the RM-type visual target than that under the presentation of the SH-type. Hence, the reason why there was no significant difference between WO- and RM-type visual tasks in present study may due to the small sample size.

The path length, relative frequency, and amplitudes were not affected by the presentation of the RM-type visual target, which induce saccadic eye movements, during BP standing. It has also been reported that the effect of saccade on the root mean square in the ML direction is smaller than that of smooth pursuit, and the significant improvement in upright stability is not observed due to the difference of experimental conditions [10]. Although the stance position is the same as previous reports, the frequency of saccades, the size of visual field, and the characteristics of participants (gender, age, and muscle activity etc) are varied in each experiment [11, 28]. Therefore, these factors may responsible for the postural sway attenuation with the saccadic eye movement.

Smooth pursuit but not saccade is mainly induced by the presentation of visual targets in randomized, horizontally sinusoidal, and vertically sinusoidal motion, and increases the postural instability [16]. Although the SH-type visual task contained both smooth pursuit and saccade, the COP displacement was also increased by this visual task. By contrast, it is reported that total displacement, sway area, and mean sway amplitude of trunk are decreased by the presentation of a similar visual task [13]. Although it is difficult to explain the inconsistent results regarding the effects of similar visual tasks on postural control, several factors may influence the postural control during visual tasks. The first is the environment of measurement space. Human receives visual inputs from both the central (±5 degrees) and peripheral (±100 degrees) regions [29]. A previous work was carried out in a fully illuminated space [13]. On the other hand, our experiments were carried out in a light-controlled space with black wall (less than 5 lux) to exclude the effect of visual information including the peripheral vision. It is

reported that the visual information presented in the peripheral field has a greater impact on postural control than that presented in the central one [30, 31]. Taken together, the difference of visual information from the peripheral region based on the illuminance of measurement space may explain the inconsistent results during smooth pursuit visual tasks. The second is the type of visual targets, such as color, shape, and size. For instance, it was reported that the eye movement was mostly composed of large saccade rather than smooth pursuit before training, when a small spot was used as the tracking visual target in monkey [32]. Hence, it is suggested that the patterns of eye movement during the visual task are different. The third is the differences in evaluation methods for postural sway. It is previously reported that the amplitude of the center of mass (COM) decreases with increasing the translation frequency of support surface. While, the amplitude of the COP increases with increasing the translation frequency of support surface [33]. In addition, the motion patterns of COP and COM are quite differed during steady-state walking trials [34]. Other group also reported that the displacement of the COP was larger than that of center of gravity at the initiation of sit-to-walk task [35]. Taken together, the difference of parameters used to evaluate postural stability also may explain the inconsistent results during smooth pursuit visual tasks.

## The effects of the bases of support on the postural stability during the performance of eye movement tasks

In the present study, we revealed that the amplitude of COP displacement was larger during UP standing than during BP standing. Consistent with this, we and other groups previously reported the same results using independently recruited participants [21, 36]. In addition, it was reported that the mean amplitude of COP displacement was larger in the tandem stance than that in the parallel stance under the eyes-open condition [37]. Moreover, the amplitude of the COP movement was larger while standing on the soft and narrow supports than while standing on the hard support [38]. These results indicate that the amplitude of COP displacement is closely associated with the difficulty of postural tasks. Interestingly, we discovered that the impact of smooth pursuits on the amplitude of COP displacement was larger during UP standing than during BP standing. It was previously reported that the increase in the COP mean velocity caused by eye closure was larger on the foam supporting surface than on the hard surface [39]. In addition, it was recently reported that the deterioration of postural stability while texting was larger in the tandem stance than in the normal stance [40]. Moreover, the increase in the mean velocity of the head caused by the presentation of horizontal and vertical visual targets was more remarkable in the feet apart base condition than in the semi-tandem base condition [41]. These results indicate that both afferent visual inputs and efferent eye movements have a large interfering effect on postural stability during difficult tasks.

## The effects of eye movement on the amplitudes of COP displacement

We found that the amplitudes of COP displacement were increased by horizontal eye movement during UP standing. Two types of conjugate eye movements–saccades and smooth pursuits–have been reported to affect postural sway. Saccades increased the frequency and decreased the amplitude of COP displacement [11, 28]. The mean amplitude of COP displacement is decreased by predictable and unpredictable saccades in both young and older people [42, 43]. In contrast, several controversial effects of smooth pursuit on the power spectrum density of the COP displacement have been reported. For instance, smooth pursuits increase the power spectral density of body sway [17]. However, recent studies have reported that the amplitude, but not the frequency, of trunk sway is attenuated by smooth pursuit, as in the case of saccadic eye movements [13].

Similar to the impacts of smooth-pursuit eye movements on postural control, both positive and negative interfering effects have also been reported during postural and cognitive dual-task paradigms. For instance, the amplitude of COP displacement was attenuated by search tasks in young adults [44, 45]. We and others recently reported that amplitudes of COP displacement were attenuated by N-back cognitive tasks [21, 46]. In contrast, the amplitude of COP displacement was increased by counting backward tests in young adults during the upright stance [47]. Other groups have reported that backward digit span testing and the Stroop task did not affect the amplitudes of postural sway in young adults and adolescents [48, 49]. These controversial results have been attributed to the difficulty and variety of the tasks involved [48, 50].

Difficulty and variety in cognitive tasks were also correlated with gaze displacement. The gaze displacement is larger during difficult search tasks than during easy ones [51]. It has been reported that the COP movement is larger during search (difficult) tasks than during stationary gaze (easy) tasks [52]. Interestingly, a recent study showed that cognition does not interfere with the relationship between eye movements and postural control [53]. Taken together, these results suggest that the patterns of smooth-pursuit eye movements are important for the differential effects of cognitive tasks on difficult postural control.

## The effects of similarity between eye and COP movements on the postural control

We demonstrated that the similarity between horizontal eye movement and COP displacement in the ML direction was preferentially increased during UP standing rather than during BP standing. In addition, the decrease of similarity was attenuated the displacement of the COP. Although the coherence between gaze and COP movements during quiet standing has not been revealed, it has been well-studied using voluntary adjustment of the COP on moving visual targets. For instance, postural sway can easily track the sine and Lorenz motions of visual targets but not Brownian stimulus motion [54]. The coherence between the COP and visual target was higher for slow-moving targets than for faster ones [55]. In addition, the coherence between the COP and visual target was higher in the unstable stance than in the stable stance [56]. These results indicate that the similarity between regular smooth-pursuit eye movements and COP displacement can easily increase during unstable stance and closely related to the postural instability.

On the other hand, the COP displacement is smaller during BP standing than during UP standing. By contrast, the eye movement under the presentation of the SH-type visual target is consistently larger than that under the presentation of the WO-type during both BP and UP standing. Thus, the warping distance between gaze and COP motions under the presentation of the SH-type visual target was larger than that under the presentation of the WO-type during BP standing. Despite the loss of similarity between eye and COP movements, it was surprised that the path length of COP displacement was increased by the presentation of SH-type visual target during BP standing. Therefore, it is suggested that the enhancement of COP displacement during BP standing is caused by other factors except for the similarity between eye and COP motions. Upright posture is compensated by several biofeedback such as visual, auditory, and vibrotactile information [57]. In addition, we and other group reported that cortical activation was promoted by the postural compensation under the difficult conditions [21, 58]. Therefore, the rectilinear and uniformly eye movement may impair the cortical motor commands related to postural compensation during both BP and UP standing.

Moreover, it was reported that there were no significant differences in the COP-visual target coherence between the AP and ML directions during voluntary adjustment of the COP on

moving visual targets [54]. This indicates that the susceptibility of COP displacement did not differ in the AP and ML directions when the participants were instructed to prioritize the postural tasks. However, the amplitude of COP displacement was strongly enhanced in the ML direction owing to the increase of similarity between eye and COP movements, and it was significantly attenuated by the decrease of similarity in the present study. Our results indicate that the displacement of the COP is more susceptible in the ML direction than in the AP direction during unstable stances when healthy young adults are instructed to prioritize tracking moving visual targets. Interestingly, the changes in relative percentage and amplitude of middle-frequency bandwidth were different between the AP and ML directions during visual tasks, suggesting that different mechanisms are involved in the postural control along the AP and ML directions.

The control of balance has been modeled as the classical inverted pendulum during quiet standing [59]. However, other groups indicate that the effect of eye closure on the root mean square deviation and mean velocity of the COP is opposite in the AP and ML directions, when the knees, hips and trunk were immobilized [60]. Therefore, the simple mechanical level using the inverted pendulum model pivoted at the ankle joint dominantly refers to the COP displacement in the AP direction, and other theoretical models considering a periodic loading of the right and left feet are recently proposed in the ML direction [61, 62]. Since upright standing is intricately regulated by both mechanical and neurological mechanisms, it is suggested that the displacement of the COP in the ML direction is more neurologically regulated than that in the AP direction.

## Limitations of this study

This study has several limitations. First, the sample size ($n$ = 14) was small because it was designed as a pilot study and the sample size was determined based on a previous report. Therefore, further research with a larger sample size is warranted. Second, the characteristics of the participants, including sex and ethnicity, were limited. Therefore, the generalizability of our results should be noted. Third, the measurement error of the eye-tracking methods was approximately 2 ˚ in the x- and y-axes owing to technical limitations. Fourth, the effects of negligible visual inputs included in peripheral vision on postural control, even though all measurements were performed in a shaded space. Fifth, a control condition of visual target motion on the retina with the eyes is still missing, in order to exclude the influence of visual motion per se rather than the smooth pursuit tracking of targets.

## Supporting information

**S1 Table. All data underlying the findings described in this study.**
(XLSX)

**S1 Fig. The standard deviations of COP displacement in the AP and ML directions.** (**A**) The standard deviations of COP displacement in the anteroposterior (AP) direction under the presentation of the WO-, RM-, SH-, and EL-type visual targets during BP (orange) and UP (green) standing. (**B**) The standard deviations of COP movement in the mediolateral (ML) direction under the presentation of the WO-, RM-, SH-, and EL-type visual targets during BP (orange) and UP (green) standing. (**C, D**) The standard deviations of COP movement in the anteroposterior (AP, C) and mediolateral (ML, D) directions under the presentation of the SH- and ulSH-type visual targets. The box plots represent the median, first and third quartiles (boxes), and fifth and 95th percentiles (whiskers). The number of participants: $n$ = 14. Statistical differences were analyzed using Friedman's analysis of variance followed by multiple

Wilcoxon's signed-rank test with Bonferroni correction. Abbreviations: AP, anteroposterior; BP, bipedal; ML, mediolateral; SD, standard deviation; UP, unipedal. Statistical significance is indicated by asterisks: $^*$ $P < 0.00833$, $^{**}$ $P < 0.00167$.
(TIF)

## Author Contributions

**Conceptualization:** Hikaru Nakahara, Tomohiro Ohgomori.

**Data curation:** Rukia Nawata, Ryota Matsuo, Tomohiro Ohgomori.

**Formal analysis:** Hikaru Nakahara, Rukia Nawata, Ryota Matsuo, Tomohiro Ohgomori.

**Funding acquisition:** Tomohiro Ohgomori.

**Investigation:** Tomohiro Ohgomori.

**Methodology:** Hikaru Nakahara, Rukia Nawata, Ryota Matsuo, Tomohiro Ohgomori.

**Project administration:** Tomohiro Ohgomori.

**Resources:** Tomohiro Ohgomori.

**Software:** Tomohiro Ohgomori.

**Supervision:** Tomohiro Ohgomori.

**Validation:** Hikaru Nakahara, Rukia Nawata, Ryota Matsuo, Tomohiro Ohgomori.

**Visualization:** Hikaru Nakahara, Rukia Nawata, Ryota Matsuo, Tomohiro Ohgomori.

**Writing – original draft:** Tomohiro Ohgomori.

**Writing – review & editing:** Tomohiro Ohgomori.

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
