## [Decision Letter · Decision Letter 0]

22 Aug 2022

PONE-D-22-19092Deterioration of postural control due to the synchronization between center of pressure and smooth pursuit eye movementsPLOS ONE

Dear Dr. Ohgomori,

Thank you for submitting your manuscript to PLOS ONE. After careful consideration, we feel that it has merit but does not fully meet PLOS ONE’s publication criteria as it currently stands. Therefore, we invite you to submit a revised version of the manuscript that addresses the points raised during the review process. While the reviewers find the results to be of potential interest, they also have major concerns requiring your attention. In particular, they ask to provide missing details in the methods and results, more comprehensive description of some algorithms, smooth pursuit vs saccades, and add more discussion on the potential consequences of visual stimuli and eye movements and possible neurophysiological mechanisms of the relation between eye movements and body sway (see their comments).

We look forward to receiving your revised manuscript.

Kind regards,

Yury Ivanenko

Academic Editor

PLOS ONE

Journal Requirements:

"This work was supported in part by the Grants-in-Aid for Scientific Research (KAKENHI) from the Japan Society for the Promotion of Science (20K07738 to T.O.) and the Research Foundation for Dementia of Osaka."

Reviewers' comments:

Reviewer's Responses to Questions

**Comments to the Author**

1. Is the manuscript technically sound, and do the data support the conclusions?

Reviewer #1: Partly

Reviewer #2: Partly

2. Has the statistical analysis been performed appropriately and rigorously? 

Reviewer #1: Yes

Reviewer #2: Yes

3. Have the authors made all data underlying the findings in their manuscript fully available?

Reviewer #1: No

Reviewer #2: Yes

4. Is the manuscript presented in an intelligible fashion and written in standard English?

Reviewer #1: Yes

Reviewer #2: Yes

5. Review Comments to the Author

Reviewer #1: The authors present an interesting study on the interaction of tracking eye movements and body sway. Although this endeavour is not novel, the present approach and the results are of potential interest to the neuroscientific community.

I have the following concerns regarding the manuscript:

In the Abstract, the authors mention both smooth pursuit and saccades as eye movement types used to track objects. Saccades, however, are used to refixate (and not to track) targets in a visual scene. Tracking is only accomplished by the smooth pursuit system (and with the optokinetic system with regard to movements of large parts of the visual field)

In introduction, the term “movement activity of the COP” is used without explanation. Most posturography studies refer to path length, sway path, sway area, sway velocity, RMS etc. Is “movement activity” analogous to these terms? Is it a combo index of path length and sway frequency?

The participants had “no history of orthopaedic diseases”. A history of neurological diseases might have been even more important in this respect.

It is still unclear why the authors used dynamic time warping analysis to examine the synchronisation between eye movements and COP fluctuations. I am not familiar with this method, and I would appreciate a more comprehensive description of this algorithm, along with a comparison with classical methods such as crosscorrelation or coherence (or non-linear methods such as mutual information)

Eye movements (gaze) should be given in degrees of visual angle rather than in pixels or cm (both in the text and in the plots), in order to gain an idea about the true magnitude of eye displacements used in this study.

The authors comment on the analogy of (sinusoidal) smooth pursuit and COP sway based on the classical inverted pendulum model. However, the latter refers only to AP sway. In contrast, ML sway (which is found to be correlated with smooth pursuit in the present study) is not explained by the inverted pendulum equilibrium concept. It is rather a periodic loading-unloading of the right and left foot.

There is no discussion on possible neurophysiological and neurobiological mechanisms of the (causal?) relation between eye movements and ML sway. The authors should elaborate on this.

A control condition of visual target movement moving on the retina with the eyes still is missing, in order to exclude the influence of visual motion per se rather than smooth eye tracking of the target. This should be mentioned in the limitations of the study.

Reviewer #2: The current study examined the effects of eye movements (a form of smooth pursuit and saccades) on upright standing. Bipedal and unipedal standing conditions were examined while no eye movements, saccades, or smooth pursuit eye movements were recorded. The authors report that COP displacements were larger with smooth pursuit eye movements, and the eye movements and COP displacements were synchronized during the smooth pursuit condition. While this research question is not novel, there are some interesting results. However, there is missing detail in the methods and the discussion appears incomplete.

Major concerns

- The authors have included a saccade condition, called the RM condition. The results of this condition show no difference to the “no visual target” (WO) condition, yet there is consistent evidence described in the intro and discussion that saccades significantly impact postural control. The authors do not discuss the results (or lack thereof) of this condition, even though it goes against most of the cited research. The authors are encouraged to expand their discussion to include the saccades results.

- There are details missing in the methods section. Specifically, there is no description of how the COP data is quantified. Movement activity is reported, but it is not clear what summary measure was used. Was root means square used, or perhaps path length? Was the data bias removed prior to calculating the amplitude (this will matter for certain amplitude measures)? Without these details, it is difficult to interpret the results. The authors are encouraged to include all necessary details for data collection and analysis.

- The smooth pursuit task is not solely a smooth pursuit task as there appears to be a saccade component within it. If I understand the SH task correctly, the participants view a target that starts on the right and translates to the left at a constant speed. The target then returns to the right side and continues to move leftward. Therefore, there is a saccade to bring the eyes from the left side of the screen to the right. Given this task is not solely a smooth pursuit task, the authors are encouraged to reclassify their description of the task and discuss the potential consequences of this task compared to a true smooth pursuit task.

- The authors state (in the title as well as throughout the manuscript) that postural control deteriorates due to the synchronization of COP and smooth pursuit eye movements. However, the results do not support this. For example, the authors report DTW distance was extended during BP yet decreased during UP for the SH compared to WO conditions. How can deterioration occur for SH conditions with opposite effects across postural tasks? The authors are encouraged to revise the discussion.

Minor concerns

- Figure 5 E: The label has a typo. DTW ratio (upSH/pSH) should be DTW ratio (upSH/SH).

- Consider changing one of the acronyms for unpredictable or unipedal, as both use “UP”.

- Page 15-16 line 358-363: This is an example where the authors write these results as if they were from the current study. However, there was no tandem stance condition or soft and narrow condition. Please consider revising this section (and all other sections with this style) to more accurately state these as referenced results.

- Page 5 line 97. How was visual acuity tested and identified as >1.0?

- Why were the participants tested 1 week apart (and then another week apart for the upSH condition)?

- Page 9 line 193: Is there a reference required for this section?

6. PLOS authors have the option to publish the peer review history of their article (what does this mean?). If published, this will include your full peer review and any attached files.

Reviewer #1: No

Reviewer #2: No

---

## [Author Response · Author response to Decision Letter 0]

7 Sep 2022

Reply to comments

Response to Academic Editor’s comments

Comment 1: Please ensure that your manuscript meets PLOS ONE's style requirements, including those for file naming. The PLOS ONE style templates can be found at 

Reply: We have ensured that our manuscript meets the required style including file naming.

Comment 2: Thank you for stating in your Funding Statement: 

"This work was supported in part by the Grants-in-Aid for Scientific Research (KAKENHI) from the Japan Society for the Promotion of Science (20K07738 to T.O.) and the Research Foundation for Dementia of Osaka."

Reply: According to comments of this and editorial office, we have deleted the sentence related to funding statement in our manuscript, and have added the sentence within cover letter as follows: 

This work was supported by the Grants-in-Aid for Scientific Research (KAKENHI) from the Japan Society for the Promotion of Science (20K07738 to T.O.) and the Research Foundation for Dementia of Osaka. The funders had no role in study design, data collection and analysis, decision to publish, or preparation of the manuscript. There was no additional external funding received for this study.

Comment 3: PLOS requires an ORCID iD for the corresponding author in Editorial Manager on papers submitted after December 6th, 2016. Please ensure that you have an ORCID iD and that it is validated in Editorial Manager. To do this, go to ‘Update my Information’ (in the upper left-hand corner of the main menu), and click on the Fetch/Validate link next to the ORCID field. This will take you to the ORCID site and allow you to create a new iD or authenticate a pre-existing iD in Editorial Manager. Please see the following video for instructions on linking an ORCID iD to your Editorial Manager account: https://www.youtube.com/watch?v=_xcclfuvtxQ

Reply: According to the comment, the corresponding author has ensured an ORCID iD and validated in Editorial Manager. 

Comment 4: Please include captions for your Supporting Information files at the end of your manuscript, and update any in-text citations to match accordingly. Please see our Supporting Information guidelines for more information: http://journals.plos.org/plosone/s/supporting-information.

Reply: The authors apologize for missing description. According to the comment, we have added the caption for the Supporting Information at the end of revised manuscript as follows:

(page 31, line 689-line 705)

Supporting Information

S1 Table All data underlying the findings described in this study. 

Figure S1 The standard deviations of COP displacement in the AP and ML directions. (A) The standard deviations of COP displacement in the anteroposterior (AP) direction under the presentation of the WO-, RM-, SH-, and EL-type visual targets during BP (orange) and UP (green) standing. (B) The standard deviations of COP movement in the mediolateral (ML) direction under the presentation of the WO-, RM-, SH-, and EL-type visual targets during BP (orange) and UP (green) standing. (C, D) The standard deviations of COP movement in the anteroposterior (AP, C) and mediolateral (ML, D) directions under the presentation of the SH- and ulSH-type visual targets. The box plots represent the median, first and third quartiles (boxes), and fifth and 95th percentiles (whiskers). The number of participants: n = 14. Statistical differences were analyzed using Friedman’s analysis of variance followed by multiple Wilcoxon’s signed-rank test with Bonferroni correction. Abbreviations: AP, anteroposterior; BP, bipedal; ML, mediolateral; SD, standard deviation; UP, unipedal. Statistical significance is indicated by asterisks: * P < 0.00833, ** P < 0.00167.

In addition, we have cited in the revised manuscript. 

 

Response to Reviewers’ Comments

General comments

1. Is the manuscript technically sound, and do the data support the conclusions?

Reviewer #1: Partly

Reviewer #2: Partly

Reply: We have totally corrected the manuscript according to reviewers’ specific comments described below.

2. Has the statistical analysis been performed appropriately and rigorously?

Reviewer #1: Yes

Reviewer #2: Yes

Reply: We appreciated for the positive evaluation of our statistical analysis.

3. Have the authors made all data underlying the findings in their manuscript fully available?

Reviewer #1: No

Reviewer #2: Yes

Reply: The data and summary of statistics were provided as supporting information and Table, respectively. In addition, the values of median, first and third quartiles have been added in Supporting Information.

4. Is the manuscript presented in an intelligible fashion and written in standard English?

Reviewer #1: Yes

Reviewer #2: Yes

Reply: We appreciated for the positive evaluation of typographical and grammatical points.

Specific comments

Reviewer #1 

Comment 1: In the Abstract, the authors mention both smooth pursuit and saccades as eye movement types used to track objects. Saccades, however, are used to refixate (and not to track) targets in a visual scene. Tracking is only accomplished by the smooth pursuit system (and with the optokinetic system with regard to movements of large parts of the visual field)

Reply: We appreciate this helpful comment. According to the comment, we have deleted the description “track visual targets”, and have corrected as follows:

(page 2, line 23-24) There are two types of efferent and conjugate eye movements: saccades and smooth pursuits.

(page 3, line 58-60) There are two main types of efferent and conjugate eye movements: saccades and smooth pursuits.

(page 20, line 467-468) Two types of conjugate eye movements–saccades and smooth pursuits–have been reported to affect postural sway.

Comment 2: In introduction, the term “movement activity of the COP” is used without explanation. Most posturography studies refer to path length, sway path, sway area, sway velocity, RMS etc. Is “movement activity” analogous to these terms? Is it a combo index of path length and sway frequency?

Reply: The authors apologize for the misleading description. We have deleted the term “movement activity of the COP” in the revised manuscript and clarified the meaning of this term through the revised manuscript. 

Comment 3: The participants had “no history of orthopaedic diseases”. A history of neurological diseases might have been even more important in this respect.

Reply: We appreciate this helpful comment. According to the comment, we additionally interviewed all participants and confirmed that they had no history of neurological diseases. Therefore, we have added the sentence as follows:

(page 4, line 92-page 5, line 93) The research subjects were healthy and had no history of orthopedic and neurological diseases.

Comment 4: It is still unclear why the authors used dynamic time warping analysis to examine the synchronisation between eye movements and COP fluctuations. I am not familiar with this method, and I would appreciate a more comprehensive description of this algorithm, along with a comparison with classical methods such as crosscorrelation or coherence (or non-linear methods such as mutual information)

Reply: The authors apologize for the missing description. As the Reviewer #1 points out, several methods, such as cross-correlation and wavelet coherence analyses were classically used to reveal the similarity between two waveforms. According to this comment, we have added the comprehensive description of the DTW algorithm and the reason why the author adopted this method to examine the similarity between the gaze and COP movements as follows:

(page 9, line 199-206) There are several methods to measure the similarity between two time-series data, including cross-correlation and wavelet coherence analyses (Andrea et al., 2021). However, the frequency bandwidths of the COP displacement differ from those of eye movements. The dynamic time warping (DTW) method is possible to quantify the similarity of two time-series data with non-linear extension and contraction allowed, even though the frequency and the number of datasets are different. Therefore, we used the DTW analysis to measure the similarity between two temporal sequences; that is, the displacement of the COP and eye movement (Li et al., 2021).

In addition, we have added the citation about this algorithm, according to this and Reviewer #2 comment.

(page 9, line 206-209) We minimized the distance between the two temporal sequences using the DTW package in R software without band filters (Sakoe-Chiba and Itakura), because it was impossible to estimate the suitable window size for matching the COP and eye movements (Giorgino, 2009).

Moreover, we have deleted the term “synchronization” and have corrected to the term “increase of similarity” in the revised manuscript, because the similarity but not synchronicity was evaluated by the DTW analysis.

Comment 5: Eye movements (gaze) should be given in degrees of visual angle rather than in pixels or cm (both in the text and in the plots), in order to gain an idea about the true magnitude of eye displacements used in this study.

Reply: We appreciate this helpful comment. According to the comment, we have totally modified Figure 4 and 5, and the representative eye movement data was shown in degrees of visual angle. In addition, we have totally corrected the term “cm” and “pixel” related to eye movement as follows:

(page 7, line 158-159) We made the blue square move linearly to the left (12.6 °/s) and disappear at the left edge of the grey-colored grid area (Fig. 1D).

(page 8, line 166-167) We then made the blue square move linearly (12.6 °/s) and turned toward unpredictable directions at the center of the grey-colored grid area.

(page 9, line 193-196) When the subject was standing with the 27-inch monitor set at a 60 cm distance in front, the measurement error of the eye-tracking method was approximately 2 ° in the x- and y-axes owing to a limitation of the measurement, which was primarily included in the central vision

(page 24, line 553-555) Third, the measurement error of the eye-tracking methods was approximately 2 ° in the x- and y-axes owing to technical limitations.

(page 25, line 575) In the case of SH, the movement of blue square was linear at 12.6 °/s along the x-axis.

Comment 6: The authors comment on the analogy of (sinusoidal) smooth pursuit and COP sway based on the classical inverted pendulum model. However, the latter refers only to AP sway. In contrast, ML sway (which is found to be correlated with smooth pursuit in the present study) is not explained by the inverted pendulum equilibrium concept. It is rather a periodic loading-unloading of the right and left foot.

There is no discussion on possible neurophysiological and neurobiological mechanisms of the (causal?) relation between eye movements and ML sway. The authors should elaborate on this.

Reply: We appreciate this helpful comment. According to this comment, we have added the description about the difference of theoretical models in the AP and ML directions and have additionally discussed about the possible neurobiological mechanisms of the relationship between COP and eye movements in the ML direction as follows:

(page 23, line 533-546) 

Interestingly, the changes in relative percentage and amplitude of middle-frequency bandwidth were different between the AP and ML directions during visual tasks, suggesting that different mechanisms are involved in the postural control along the AP and ML directions.

The control of balance has been modeled as the classical inverted pendulum during quiet standing (Gage et al., 2004). However, other groups indicate that the effect of eye closure on the root mean square deviation and mean velocity of the COP is opposite in the AP and ML directions, when the knees, hips and trunk were immobilized (Freitas et al., 2009). Therefore, the simple mechanical level using the inverted pendulum model pivoted at the ankle joint dominantly refers to the COP displacement in the AP direction, and other theoretical models considering a periodic loading of the right and left feet are recently proposed in the ML direction (Rusaw et al., 2021; Winter, 1995). Since upright standing is intricately regulated by both mechanical and neurological mechanisms, it is suggested that the displacement of the COP in the ML direction is more neurologically regulated than that in the AP direction.

In addition, we also discussed about the possible neurological mechanisms of the relation between eye movement and postural compensation as follows:

(page 22, line 517-522) Upright posture is compensated by several biofeedback such as visual, auditory, and vibrotactile information (Horak, 2010). In addition, we and other group reported that cortical activation was promoted by the postural compensation under the difficult conditions (St George et al., 2021; Sugihara et al., 2021). Therefore, the rectilinear and uniformly eye movement may impair the cortical motor commands related to postural compensation during both BP and UP standing.

Comment 7: A control condition of visual target movement moving on the retina with the eyes still is missing, in order to exclude the influence of visual motion per se rather than smooth eye tracking of the target. This should be mentioned in the limitations of the study.

Reply: We appreciate this helpful comment. According to the comment, we have added the sentence in the “Limitations of this study” section as follows: 

(page 24, line 556-559) Fifth, a control condition of visual target motion on the retina with the eyes is still missing, in order to exclude the influence of visual motion per se rather than the smooth pursuit tracking of targets.

Reviewer #2

Major concerns

Comment 1: The authors have included a saccade condition, called the RM condition. The results of this condition show no difference to the “no visual target” (WO) condition, yet there is consistent evidence described in the intro and discussion that saccades significantly impact postural control. The authors do not discuss the results (or lack thereof) of this condition, even though it goes against most of the cited research. The authors are encouraged to expand their discussion to include the saccades results.

Reply: We appreciate this critical comment. As the Reviewer #2 points out, it is reported that the saccadic eye movement has a significant impact on postural control (Rey et al., 2008; Rodrigues et al., 2013; Rougier & Garin, 2007). According to this comment, we additionally discussed about the effects of eye movement tasks on the postural control in newly created subsection as follows:

(page 17, line 390-403) 

The effects of eye movement tasks on the postural control 

It was previously reported that the saccadic eye movement significantly attenuated postural control (Rey et al., 2008; Rodrigues et al., 2013; Rougier & Garin, 2007). In this study, we revealed that the path length of the COP displacement under the presentation of the RM-type visual target tend to be smaller than that under the presentation of the WO-type one during UP standing (P-value = 0.0419 in a direct comparison between two groups) but not BP standing. In addition, the amplitude of the high-frequency bandwidth in the ML direction under the presentation of the RM-type visual target also tend to be smaller than that under the presentation of the WO-type one (P-value = 0.0245 in a direct comparison between two groups) but not BP standing. However, the change in the path length of the COP displacement and the amplitude of high-frequency bandwidth in the ML direction were smaller under the presentation of the RM-type visual target than that under the presentation of the SH-type one. Hence, the reason why there was no significant difference between WO- and RM-type visual tasks in present study may due to the small sample size. 

Another critical point is that our findings do not align with previous data showing improvements in postural stability using saccadic visual tasks during BP standing. We also discussed about these controversial results as follows: 

(page 17, line 404-page 18, line 412) 

The path length, relative frequency, and amplitudes were not affected by the presentation of the RM-type visual target, which induce saccadic eye movements, during BP standing. It has also reported that the effect of saccade on the root mean square in the ML direction is smaller than that of smooth pursuit, and the significant improvement in upright stability is not observed due to the difference of experimental conditions (Thomas et al., 2016). Although the stance position is the same as previous reports, the frequency of saccades, the size of visual field, and the characteristics of participants (gender, age, and muscle activity etc) are varied in each experiment (Aguiar et al., 2015; Rodrigues et al., 2013). Therefore, these factors may responsible for the postural sway attenuation with the saccadic eye movement.

Comment 2: There are details missing in the methods section. Specifically, there is no description of how the COP data is quantified. Movement activity is reported, but it is not clear what summary measure was used. Was root means square used, or perhaps path length? 

Reply: The authors apologize for the misleading description. According to this and Reviewer #1’s comment, we have deleted the term “movement activity of the COP” in the revised manuscript and clarified the meaning of this term through the revised manuscript.

Comment 3: Was the data bias removed prior to calculating the amplitude (this will matter for certain amplitude measures)? Without these details, it is difficult to interpret the results. The authors are encouraged to include all necessary details for data collection and analysis.

Reply: We apologized for the missing description. According to this comment, we have calculated the standard deviations of the COP displacement in the AP and ML directions (Supplementary Figure 1). Based on the distributions of standard deviation, the data that is 1.5 × interquartile range (IQR) greater than the third quartile and the data that is 1.5 × IQR less than the first quartile were excluded. Then, we have recalculated the amplitudes of COP displacement. 

Although the similar statistical results were obtained shown in above, the excluded data could not be defined as outliers due to the small sample size in present study. Therefore, we have added the supplementary figure, data, and the detail description for data analysis as follows:

(page 8, line 172-179 in Methods section) Prior to calculating amplitude, we examined the distributions of standard deviation (SD) of the COP displacement in the AP and ML directions (Fig. S1). There were several data which was more than 1.5 interquartile ranges below the first quartile or above the third quartile. However, these data could not be designated as outliers due to the small sample size in present study. Therefore, the temporal data in the AP and ML directions obtained from all participants during BP and UP standing (60 s) were changed to frequencies using Bluestein’s fast Fourier transformations, as reported previously (Sugihara et al., 2021). These signals were low pass filtered with a cut-off at 3 Hz based on the previous report (Loram et al., 2006).

(page 11, line 260-page 12, line 269 in Results section) Prior to changing to frequencies using fast Fourier transformation, the changes in SD of temporal displacement of the COP in the AP and ML directions were examined (Fig. S1). The SD in the AP direction remained unaltered among the four groups during BP standing (χ2 3,39 = 1.89, P = 0.596, Fig. S1A, S1 Table). The SD in the AP direction was slightly, but not significantly, altered (χ2 3,39 = 8.31, P = 0.0399, Fig. S1A, S1 Table, Table 1). The SD in the ML direction also remained unaltered among the four groups during BP standing (χ2 3,39 = 3.26, P = 0.354, Fig. S1A, S1 Table). By contrast, The SD in the ML direction was significantly increased by the presentation of SH-type visual target (χ2 3,39 = 20.9, P = 0.00011, Fig. S1A, S1Table, Table 1). These data indicate that body sway was increased in the ML direction by the presentation of SH-type visual target.

(page 16, line 374-378 in Results section) Prior to changing to frequencies using fast Fourier transformation, the change in SD of temporal displacement of the COP in the AP and ML directions were examined (Fig. S1). The SD in the AP direction was increased by the presentation of ulSH-type visual target (Z = -2.04, P = 0.0419, Fig. S1C). By contrast, the SD in the ML direction did not differ between two groups (Z = -1.35, P = 0.194, Fig. S1D).

Comment 4: The smooth pursuit task is not solely a smooth pursuit task as there appears to be a saccade component within it. If I understand the SH task correctly, the participants view a target that starts on the right and translates to the left at a constant speed. The target then returns to the right side and continues to move leftward. Therefore, there is a saccade to bring the eyes from the left side of the screen to the right. Given this task is not solely a smooth pursuit task, the authors are encouraged to reclassify their description of the task and discuss the potential consequences of this task compared to a true smooth pursuit task.

Reply: We appreciate the critical comment. As the Reviewer #2 points out, the large amount of smooth pursuit and the small amount of saccade were included in the SH-type visual task. By contrast, the smooth pursuit was rarely included in the RM-type visual task. According to this comment, the visual tasks were reclassified in Methods section as follows:

(page 7, line 155-156) The RM-type visual task mainly induced the saccadic eye movement and rarely induced the smooth pursuit.

(page 7, line 160-162) The SH-type visual task mainly induced the smooth-pursuit eye movement and partially induced the saccade, when the eyes were moved from the left side of the screen to the right.

In addition we additionally discussed about the potential consequence of SH-type visual task compared to smooth pursuit tasks reported previously; i.e. the same as our task and true smooth pursuit tasks as follows: 

(page 18, line 413-page 19, line 442 in Discussion section) Smooth pursuit but not saccade is mainly induced by the presentation of visual targets in randomized, horizontally sinusoidal, and vertically sinusoidal motion, and increases the postural instability (Kim et al., 2016). Although the SH-type visual task contained both smooth pursuit and saccade, the COP displacement was also increased by this visual task. By contrast, it is reported that total displacement, sway area, and mean sway amplitude of trunk are decreased by the presentation of a similar visual task (Rodrigues et al., 2015). Although it is difficult to explain the inconsistent results regarding the effects of similar visual tasks on postural control, several factors may influence the postural control during visual tasks. The first is the environment of measurement space. Human receives visual inputs from both the central (±5 degrees) and peripheral (±100 degrees) regions (Lungaro et al., 2018). A previous work was carried out in a fully illuminated space (Rodrigues et al., 2015). On the other hand, our experiments were carried out in a light-controlled space with black wall (less than 5 lux) to exclude the effect of visual information including the peripheral vision. It is reported that the visual information presented in the peripheral field has a greater impact on postural control than that presented in the central one (Amblard & Carblanc, 1980; Brandt et al., 1973). Taken together, the difference of visual information from the peripheral region based on the illuminance of measurement space may explain the inconsistent results during smooth pursuit visual tasks. The second is the type of visual targets, such as color, shape, and size. For instance, it was reported that the eye movement was mostly composed of large saccade rather than smooth pursuit before training, when a small spot was used as the tracking visual target in monkey (Botschko et al., 2018). Hence, it is suggested that the patterns of eye movement during the visual task are different. The third is the differences in evaluation methods for postural sway. It is previously reported that the amplitude of the center of mass (COM) decreases with increasing the translation frequency of support surface. While, the amplitude of the COP increases with increasing the translation frequency of support surface (Buchanan & Horak, 1999). In addition, the motion patterns of COP and COM are quite differed during steady-state walking trials (Jian et al., 1993). Other group also reported that the displacement of the COP was larger than that of center of gravity at the initiation of sit-to-walk task (Asakura & Usuda, 2013). Taken together, the difference of parameters used to evaluate postural stability also may explain the inconsistent results during smooth pursuit visual tasks.

Comment 5: The authors state (in the title as well as throughout the manuscript) that postural control deteriorates due to the synchronization of COP and smooth pursuit eye movements. However, the results do not support this. For example, the authors report DTW distance was extended during BP yet decreased during UP for the SH compared to WO conditions. How can deterioration occur for SH conditions with opposite effects across postural tasks? The authors are encouraged to revise the discussion.

Reply: We apologize for the misleading description. First, the DTW distance indicates the “similarity” but not “synchronicity” between two time-series data. According to this comment, we have totally deleted the term “synchronization” and have corrected the title as follows:

(page 1, line 1-2) Deterioration of postural control due to the increase of similarity between center of pressure and smooth-pursuit eye movements during standing on one leg.

As the Reviewer #2 points out, the path length of COP displacement was significantly increased by the presentation of SH-type visual target during BP standing, despite the loss of similarity between eye and COP motions. Therefore, we totally rewrote the last subsection in Discussion and additionally discussed about the opposite effects across postural tasks as following underlines: 

(page 21, line 494-page 22, line 522) 

The effects of similarity between eye and COP movements on the postural control

We demonstrated that the similarity between horizontal eye movement and COP displacement in the ML direction was preferentially increased during UP standing rather than during BP standing. In addition, the decrease of similarity was attenuated the displacement of the COP. Although the coherence between gaze and COP movements during quiet standing has not been revealed, it has been well-studied using voluntary adjustment of the COP on moving visual targets. For instance, postural sway can easily track the sine and Lorenz motions of visual targets but not Brownian stimulus motion (Hatzitaki et al., 2015). The coherence between the COP and visual target was higher for slow-moving targets than for faster ones (Patikas et al., 2019). In addition, the coherence between the COP and visual target was higher in the unstable stance than in the stable stance (Mademli et al., 2021). These results indicate that the similarity between regular smooth-pursuit eye movements and COP displacement can easily increase during unstable stance and closely related to the postural instability.

On the other hand, the COP displacement is smaller during BP standing than during UP standing. By contrast, the eye movement under the presentation of the SH-type visual target is consistently larger than that under the presentation of the WO-type one during both BP and UP standing. Thus, the warping distance between gaze and COP motions under the presentation of the SH-type visual target was larger than that under the presentation of the WO-type one during BP standing. Despite the loss of similarity between eye and COP movements, it was surprised that the path length of COP displacement was increased by the presentation of SH-type visual target during BP standing. Therefore, it is suggested that the enhancement of COP displacement during BP standing is caused by other factors except for the similarity between eye and COP motions. Upright posture is compensated by several biofeedback such as visual, auditory, and vibrotactile information (Horak, 2010). In addition, we and other group reported that cortical activation was promoted by the postural compensation under the difficult conditions (St George et al., 2021; Sugihara et al., 2021). Therefore, the rectilinear and uniformly eye movement may impair the cortical motor commands related to postural compensation during both BP and UP standing.

Minor concerns

Comment 6: Figure 5 E: The label has a typo. DTW ratio (upSH/pSH) should be DTW ratio (upSH/SH).

Reply: We apologize the typographical error in Figure 5E. According to this comment, we have corrected the label in Figure 5E.

Comment 7: Consider changing one of the acronyms for unpredictable or unipedal, as both use “UP”.

Reply: We apologize the misleading description. According to this comment, we have totally changed the acronyms for unpredictable to “ul” based on the previous report (Nätt et al., 2009).

Comment 8: Page 15-16 line 358-363: This is an example where the authors write these results as if they were from the current study. However, there was no tandem stance condition or soft and narrow condition. Please consider revising this section (and all other sections with this style) to more accurately state these as referenced results.

Reply: We apologize the misleading description. According to this comment, we have changed the order of sentences and corrected as follows:

(page 19, line 446-452) In present study, we revealed that the amplitude of COP displacement was larger during UP standing than during BP standing. Consistent with this, we and other groups previously reported the same results using independently recruited participants (Sugihara et al., 2021; Watanabe et al., 2018). In addition, it was reported that the mean amplitude of COP displacement was larger in the tandem stance than that in the parallel stance under the eyes-open condition. Moreover, the amplitude of the COP movement was larger while standing on the soft and narrow supports than while standing on the hard support (Krizková et al., 1993).

Comment 9: Page 5 line 97. How was visual acuity tested and identified as >1.0?

Reply: We apologize the missing the explanation of vision test. According to this comment, we have added the sentence as follows:

(page 5, line 94-95) Visual acuity was separately tested in each eye using Landolt C chart in a random order.

Comment 10: Why were the participants tested 1 week apart (and then another week apart for the upSH condition)?

Reply: We apologize the missing the description. According to the comment, we have added the reason why the participants tested 1 week apart as follows:

(page 6, line 122-125) There are two reasons why participants tested 1 week apart. One is the scheduling constraint of participants. Second is to eliminate the effect of fatigue on the COP displacement due to repeated measurement under the standing conditions based on the previous reports (Degache et al., 2020).

Comment 11: Page 9 line 193: Is there a reference required for this section?

Reply: The authors apologize the missing the description. According to the comment, we have added the reference involved in the DTW package in R and the reason why the band filters were not applied to the DTW analysis as follows: 

(page 9, line 206-210) We minimized the distance between the two temporal sequences using the DTW package in R software without band filters (Sakoe-Chiba and Itakura), because it was impossible to estimate the suitable window size for matching the COP and eye movements (Giorgino, 2009).

References

Aguiar, S. A., Polastri, P. F., Godoi, D., Moraes, R., Barela, J. A., & Rodrigues, S. T. (2015). Effects of saccadic eye movements on postural control in older adults. Psychology & Neuroscience, 8(1), 19–27. https://doi.org/10.1037/h0100352

Amblard, B., & Carblanc, A. (1980). Role of foveal and peripheral visual information in maintenance of postural equilibrium in man. Perceptual and Motor Skills, 51(3 Pt 1). https://doi.org/10.2466/pms.1980.51.3.903

Andrea, B., Atiqah, A., & Gianluca, E. (2021). Reproducible Inter-Personal Brain Coupling Measurements in Hyperscanning Settings With functional Near Infra-Red Spectroscopy. Neuroinformatics. https://doi.org/10.1007/s12021-021-09551-6

Asakura, T., & Usuda, S. (2013). Effects of directional change on postural adjustments during the sit-to-walk task. Journal of Physical Therapy Science, 25(11). https://doi.org/10.1589/jpts.25.1377

Botschko, Y., Yarkoni, M., & Joshua, M. (2018). Smooth pursuit eye movement of monkeys naive to laboratory setups with pictures and artificial stimuli. Frontiers in Systems Neuroscience, 12. https://doi.org/10.3389/fnsys.2018.00015

Brandt, T., Dichgans, J., & Koenig, E. (1973). Differential effects of central versus peripheral vision on egocentric and exocentric motion perception. Experimental Brain Research, 16(5). https://doi.org/10.1007/BF00234474

Buchanan, J. J., & Horak, F. B. (1999). Emergence of postural patterns as a function of vision and translation frequency. Journal of Neurophysiology, 81(5). https://doi.org/10.1152/jn.1999.81.5.2325

Degache, F., Serain, É., Roy, S., Faiss, R., & Millet, G. P. (2020). The fatigue-induced alteration in postural control is larger in hypobaric than in normobaric hypoxia. Scientific Reports, 10(1). https://doi.org/10.1038/s41598-019-57166-4

Freitas, P. B. de, Freitas, S. M. S. F., Duarte, M., Latash, M. L., & Zatsiorsky, V. M. (2009). Effects of joint immobilization on standing balance. Human Movement Science, 28(4). https://doi.org/10.1016/j.humov.2009.02.001

Gage, W. H., Winter, D. A., Frank, J. S., & Adkin, A. L. (2004). Kinematic and kinetic validity of the inverted pendulum model in quiet standing. Gait and Posture, 19(2). https://doi.org/10.1016/S0966-6362(03)00037-7

Giorgino, T. (2009). Computing and visualizing dynamic time warping alignments in R: The dtw package. Journal of Statistical Software, 31(7). https://doi.org/10.18637/jss.v031.i07

Hatzitaki, V., Stergiou, N., Sofianidis, G., & Kyvelidou, A. (2015). Postural Sway and Gaze Can Track the Complex Motion of a Visual Target. PLOS ONE, 10(3), e0119828. https://doi.org/10.1371/journal.pone.0119828

Horak, F. B. (2010). Postural compensation for vestibular loss and implications for rehabilitation. In Restorative Neurology and Neuroscience (Vol. 28, Issue 1). https://doi.org/10.3233/RNN-2010-0515

Jian, Y., Winter, D., Ishac, M., & Gilchrist, L. (1993). Trajectory of the body COG and COP during initiation and termination of gait. Gait and Posture, 1(1). https://doi.org/10.1016/0966-6362(93)90038-3

Kim, S.-Y., Moon, B.-Y., & Cho, H. G. (2016). Smooth-pursuit eye movements without head movement disrupt the static body balance. Journal of Physical Therapy Science, 28(4), 1335–1338. https://doi.org/10.1589/jpts.28.1335

Krizková, M., Hlavacka, F., & Gatev, P. (1993). Visual control of human stance on a narrow and soft support surface. Physiological Research, 42(4), 267–272.

Li, D., Kaminishi, K., Chiba, R., Takakusaki, K., Mukaino, M., & Ota, J. (2021). Evaluation of Postural Sway in Post-stroke Patients by Dynamic Time Warping Clustering. Frontiers in Human Neuroscience, 15. https://doi.org/10.3389/fnhum.2021.731677

Loram, I. D., Gawthrop, P. J., & Lakie, M. (2006). The frequency of human, manual adjustments in balancing an inverted pendulum is constrained by intrinsic physiological factors. Journal of Physiology, 577(1). https://doi.org/10.1113/jphysiol.2006.118786

Lungaro, P., Sjöberg, R., Valero, A. J. F., Mittal, A., & Tollmar, K. (2018). Gaze-Aware streaming solutions for the next generation of mobile VR experiences. IEEE Transactions on Visualization and Computer Graphics, 24(4). https://doi.org/10.1109/TVCG.2018.2794119

Mademli, L., Mavridi, D., Bohm, S., Patikas, D. A., Santuz, A., & Arampatzis, A. (2021). Standing on unstable surface challenges postural control of tracking tasks and modulates neuromuscular adjustments specific to task complexity. Scientific Reports, 11(1), 6122. https://doi.org/10.1038/s41598-021-84899-y

Nätt, D., Lindqvist, N., Stranneheim, H., Lundeberg, J., Torjesen, P. A., & Jensen, P. (2009). Inheritance of acquired behaviour adaptations and brain gene expression in chickens. PLoS ONE, 4(7). https://doi.org/10.1371/journal.pone.0006405

Patikas, D. A., Papavasileiou, A., Ekizos, A., Hatzitaki, V., & Arampatzis, A. (2019). Swaying slower reduces the destabilizing effects of a compliant surface on voluntary sway dynamics. PLOS ONE, 14(12), e0226263. https://doi.org/10.1371/journal.pone.0226263

Rey, F., Lê, T.-T., Bertin, R., & Kapoula, Z. (2008). Saccades horizontal or vertical at near or at far do not deteriorate postural control. Auris Nasus Larynx, 35(2), 185–191. https://doi.org/10.1016/j.anl.2007.07.001

Rodrigues, S. T., Aguiar, S. A., Polastri, P. F., Godoi, D., Moraes, R., & Barela, J. A. (2013). Effects of saccadic eye movements on postural control stabilization. Motriz: Revista de Educação Física, 19(3), 614–619. https://doi.org/10.1590/S1980-65742013000300012

Rodrigues, S. T., Polastri, P. F., Carvalho, J. C., Barela, J. A., Moraes, R., & Barbieri, F. A. (2015). Saccadic and smooth pursuit eye movements attenuate postural sway similarly. Neuroscience Letters, 584, 292–295. https://doi.org/10.1016/j.neulet.2014.10.045

Rougier, P., & Garin, M. (2007). Performing Saccadic Eye Movements or Blinking Improves Postural Control. Motor Control, 11(3), 213–223. https://doi.org/10.1123/mcj.11.3.213

Rusaw, D. F., Alinder, R., Edholm, S., Hallstedt, K. L. L., Runesson, J., & Barnett, C. T. (2021). Development of a theoretical model for upright postural control in lower limb prosthesis users. Scientific Reports, 11(1). https://doi.org/10.1038/s41598-021-87657-2

St George, R. J., Hinder, M. R., Puri, R., Walker, E., & Callisaya, M. L. (2021). Functional Near-infrared Spectroscopy Reveals the Compensatory Potential of Pre-frontal Cortical Activity for Standing Balance in Young and Older Adults. Neuroscience, 452. https://doi.org/10.1016/j.neuroscience.2020.10.027

Sugihara, Y., Matsuura, T., Kubo, Y., & Ohgomori, T. (2021). Activation of the Prefrontal Cortex and Improvement of Cognitive Performance with Standing on One Leg. Neuroscience, 477, 50–62. https://doi.org/10.1016/j.neuroscience.2021.10.004

Thomas, N. M., Bampouras, T. M., Donovan, T., & Dewhurst, S. (2016). Eye Movements Affect Postural Control in Young and Older Females. Frontiers in Aging Neuroscience, 8. https://doi.org/10.3389/fnagi.2016.00216

Watanabe, T., Saito, K., Ishida, K., Tanabe, S., & Nojima, I. (2018). Coordination of plantar flexor muscles during bipedal and unipedal stances in young and elderly adults. Experimental Brain Research, 236(5), 1229–1239. https://doi.org/10.1007/s00221-018-5217-3

Winter, D. A. (1995). Human balance and posture control during standing and walking. In Gait and Posture (Vol. 3, Issue 4). https://doi.org/10.1016/0966-6362(96)82849-9

---

## [Decision Letter · Decision Letter 1]

27 Sep 2022

PONE-D-22-19092R1Deterioration of postural control due to the increase of similarity between center of pressure and smooth-pursuit eye movements during standing on one legPLOS ONE

Dear Dr. Ohgomori,

Thank you for submitting your manuscript to PLOS ONE. After careful consideration, we feel that it has merit but does not fully meet PLOS ONE’s publication criteria as it currently stands. Therefore, we invite you to submit a revised version of the manuscript that addresses the points raised during the review process.  Please submit your revised manuscript by Nov 11 2022 11:59PM. If you will need more time than this to complete your revisions, please reply to this message or contact the journal office at plosone@plos.org. Please include the following items when submitting your revised manuscript:A rebuttal letter that responds to each point raised by the academic editor and reviewer(s). You should upload this letter as a separate file labeled 'Response to Reviewers'.A marked-up copy of your manuscript that highlights changes made to the original version. You should upload this as a separate file labeled 'Revised Manuscript with Track Changes'.An unmarked version of your revised paper without tracked changes. You should upload this as a separate file labeled 'Manuscript'.If applicable, we recommend that you deposit your laboratory protocols in protocols.io to enhance the reproducibility of your results. Protocols.io assigns your protocol its own identifier (DOI) so that it can be cited independently in the future. For instructions see: https://journals.plos.org/plosone/s/submission-guidelines#loc-laboratory-protocols. Additionally, PLOS ONE offers an option for publishing peer-reviewed Lab Protocol articles, which describe protocols hosted on protocols.io. Read more information on sharing protocols at https://plos.org/protocols?utm_medium=editorial-email&utm_source=authorletters&utm_campaign=protocols.

We look forward to receiving your revised manuscript.

Kind regards,

Yury Ivanenko

Academic Editor

PLOS ONE

Journal Requirements:

Additional Editor Comments:

The reviewers are very porsitive about the revision, and only a few minor comments remain to be addressed (see below). I hope you can fix these points soon.

Reviewers' comments:

Reviewer's Responses to Questions

**Comments to the Author**

1. If the authors have adequately addressed your comments raised in a previous round of review and you feel that this manuscript is now acceptable for publication, you may indicate that here to bypass the “Comments to the Author” section, enter your conflict of interest statement in the “Confidential to Editor” section, and submit your "Accept" recommendation.

Reviewer #1: All comments have been addressed

Reviewer #2: (No Response)

2. Is the manuscript technically sound, and do the data support the conclusions?

Reviewer #1: Yes

Reviewer #2: Yes

3. Has the statistical analysis been performed appropriately and rigorously? 

Reviewer #1: Yes

Reviewer #2: Yes

4. Have the authors made all data underlying the findings in their manuscript fully available?

Reviewer #1: No

Reviewer #2: Yes

5. Is the manuscript presented in an intelligible fashion and written in standard English?

Reviewer #1: Yes

Reviewer #2: Yes

6. Review Comments to the Author

Reviewer #1: All queries have been addressed adequately and correctly by the authors. The additions in the text as well as the corrections in the figure are satisfactory.

Reviewer #2: The re-submission of the study "Deterioration of postural control due to the increase of similarity between center of pressure and smooth-pursuit eye movements during standing on one leg" has been greatly improved. The authors have done a sufficient job at addressing most major concerns presented in the first review. I only have minor comments for revisions.

Minor Issues

- Page 6 line 120-121: How are you calculating the “amplitude of the COP displacement”? Instead of stating it multiple times in the results, (lines 260 to 261 and 374-376), consider stating this in the methods.

- Page 6 line 122-123: Consider adding the word “were” after “There are two reasons why participants”.

- Page 17: Consider removing the word “one” after XX-type throughout the discussion.

- Page 18 line 406: Consider adding the word “been” after “It has also”.

- Page 19 line 446: Consider adding the word “the” as such “In the present study”.

7. PLOS authors have the option to publish the peer review history of their article (what does this mean?). If published, this will include your full peer review and any attached files.

Reviewer #1: No

Reviewer #2: No

---

## [Author Response · Author response to Decision Letter 1]

28 Sep 2022

Reply to comments

Journal Requirements:

Reply: According to the requirement, we checked the reference list. No retracted papers were cited.

Additional Editor Comments:

The reviewers are very porsitive about the revision, and only a few minor comments remain to be addressed (see below). I hope you can fix these points soon.

Reply: We appreciated for the positive evaluation. According to reviewer’s comments, we have totally amended our manuscript. We hope that the revised manuscript is now acceptable for publication in PLoS One.

Response to Reviewers’ Comments

Reviewer #1: All queries have been addressed adequately and correctly by the authors. The additions in the text as well as the corrections in the figure are satisfactory.

Reply: We appreciated for the positive evaluation.

Reviewer #2: 

Comment 1: - Page 6 line 120-121: How are you calculating the “amplitude of the COP displacement”? Instead of stating it multiple times in the results, (lines 260 to 261 and 374-376), consider stating this in the methods.

Reply: The authors apologize for the missing description. According to comment, we have added the methodology of calculating the amplitude of the COP displacement in the Methods section as follows:

(page 8, line 181-184) The relative proportion of the area under the spectral plots of power in each frequency bandwidth was calculated. The sum of the area under the spectral plots of amplitude in each frequency bandwidth was designated as the amplitude of the COP displacement.

Comment 2: - Page 6 line 122-123: Consider adding the word “were” after “There are two reasons why participants”.

Reply: The authors apologize for the missing description. According to comment, we have corrected the sentence as follows:

(page 6, line 122-123) There are two reasons why participants were tested 1 week apart.

Comment 3: - Page 17: Consider removing the word “one” after XX-type throughout the discussion.

Reply: The authors appreciate the helpful comment. According to comment, we have totally deleted the word “one” after XX-type throughout the discussion section.

(page 17, line 396 & 399; page 22, line 511 & 514) WO-type

(page 17, line 403) SH-type

Comment 4: - Page 18 line 406: Consider adding the word “been” after “It has also”.

Reply: According to comment, we have corrected the sentence as follows:

(page 18, line 408-410) It has also been reported that the effect of saccade on the root mean square in the ML direction is smaller than that of smooth pursuit, and the significant improvement in upright stability is not observed due to the difference of experimental conditions

Comment 5: - Page 19 line 446: Consider adding the word “the” as such “In the present study”.

Reply: According to comment, we have corrected the sentence as follows:

(page 19, line 448-449) In the present study, we revealed that the amplitude of COP displacement was larger during UP standing than during BP standing.

---

## [Editor Report · Decision Letter 2]

29 Sep 2022

Deterioration of postural control due to the increase of similarity between center of pressure and smooth-pursuit eye movements during standing on one leg

PONE-D-22-19092R2

Dear Dr. Ohgomori,

We’re pleased to inform you that your manuscript has been judged scientifically suitable for publication and will be formally accepted for publication once it meets all outstanding technical requirements.

Kind regards,

Yury Ivanenko

Academic Editor

PLOS ONE
---

## [Editor Report · Acceptance letter]

3 Oct 2022

PONE-D-22-19092R2 

Deterioration of postural control due to the increase of similarity between center of pressure and smooth-pursuit eye movements during standing on one leg 

Dear Dr. Ohgomori:

I'm pleased to inform you that your manuscript has been deemed suitable for publication in PLOS ONE. Congratulations! Your manuscript is now with our production department. 

Kind regards, 

on behalf of

Dr. Yury Ivanenko 

Academic Editor

PLOS ONE